# NPCL: Neural Processes for Uncertainty-Aware Continual Learning

**Saurav Jha**
UNSW Sydney
saurav.jha@unsw.edu.au

**Dong Gong**[*]
UNSW Sydney
dong.gong@unsw.edu.au

**He Zhao**
CSIRO's Data61
he.zhao@ieee.org

**Lina Yao**
CSIRO's Data61, UNSW Sydney
lina.yao@data61.csiro.au

## Abstract

Continual learning (CL) aims to train deep neural networks efficiently on streaming data while limiting the forgetting caused by new tasks. However, learning transferable knowledge with less interference between tasks is difficult, and real-world deployment of CL models is limited by their inability to measure predictive uncertainties. To address these issues, we propose handling CL tasks with neural processes (NPs), a class of meta-learners that encode different tasks into probabilistic distributions over functions all while providing reliable uncertainty estimates. Specifically, we propose an NP-based CL approach (NPCL) with task-specific modules arranged in a hierarchical latent variable model. We tailor regularizers on the learned latent distributions to alleviate forgetting. The uncertainty estimation capabilities of the NPCL can also be used to handle the task head/module inference challenge in CL. Our experiments show that the NPCL outperforms previous CL approaches. We validate the effectiveness of uncertainty estimation in the NPCL for identifying novel data and evaluating instance-level model confidence. Code is available at https://github.com/srvCodes/NPCL.

## 1 Introduction

Continual learning (CL) aims to help deep neural networks (DNNs) learn from a stream of non-stationary tasks by retaining the previously acquired knowledge [54, 35]. To achieve this, CL agents target alleviating the *catastrophic forgetting* issue with restricted computational and memory costs [42]. This requires balancing the plasticity for new knowledge with the stability for old [37].

To handle forgetting in CL, experience replay (ER) methods [30, 6] are one effective way to train DNNs on a memory buffer with a subset of the past tasks' experiences. Other than the ER methods, many regularization-based approaches have been proposed to penalize the forgetting on the DNNs' parametric [30] or representation spaces [5, 4]. However, these may still suffer from interference due to the regularization on the entire parameter space [7]. To address this, parameter isolation methods [53, 33] define task-specific training components but are usually confined to task incremental CL setups requiring task ID during testing [47]. It is thus challenging for CL agents to maintain transferable and shareable knowledge. Lastly, a hurdle to the real-world deployment of CL agents is their inability to measure predictive uncertainties, which impacts the potential utilization of CL across various practical applications, particularly those with critical safety considerations [31].

---

[*]D. Gong is the corresponding author.

37th Conference on Neural Information Processing Systems (NeurIPS 2023).

To tackle the above issues, we propose to explore CL models using neural processes (NPs) [13, 14], a class of meta-learners that model tasks as data-generating functions from a stochastic process. NPs learn a prior over functions by marginalizing over a set of data points, or *context*, thus enabling rapid adaptation to new observations through inference on functions. Additionally, their probabilistic nature endows them with reliable uncertainty quantification capabilities [14, 27, 24, 25]. Our motivations to explore NPs for CL are thus two-fold. First, NPs exploit Bayes' theorem, which naturally enables CL through sequential posterior construction. Namely, NPs perform inference over the function space by learning context-based priors, which are updated to posteriors upon observing (additional) *targets*. Second, NPs meta-learn input correlations through a set of latent variables, which could be a key to meta-learn knowledge transfer across multiple correlated tasks. However, NPs face challenges in directly addressing CL tasks, given that (a) the reliance on a single global latent leads to suboptimal modeling of complex CL tasks where multiple correlated tasks could occur simultaneously, (b) NPs cannot directly handle the forgetting of past task correlations arising from the non-static data stream.

To address the above desiderata, we propose Neural Processes for Continual Learning (NPCL), a hierarchical latent variable model with a global latent variable to capture inter-task correlation and task-specific latent variables for finer knowledge. Fig. 1 shows the NPCL exploiting functional correlation among current and past task training samples of ER. The drift of global and past task-specific distributions away from their original forms is the major cause of forgetting in the NPCL. We thus propose to regularize the latent variables to be similar to their old forms and show the merits of regularization over typical parameter-based regularization. We then leverage the uncertainty encoded by the NPCL for the aforesaid CL challenge of task head inference. To this end, we propose using entropy as an uncertainty quantification metric (UQM). The NPCL outperforms previous probabilistic CL models

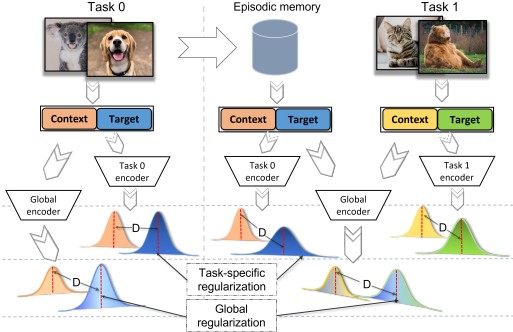

Figure 1: **Neural Processes for Continual Learning (NPCL)**: each training step involves minimizing the distance $D$ between the context-based prior and the target-based posterior, alongside regularizing the task-specific and global distributions towards their old forms.

and delivers better or comparable results than state-of-the-art deterministic CL methods, which usually have an edge over their probabilistic counterparts in terms of accuracy. Moreover, our ablations show the enhanced efficacy offered by the NPCL on continual learning settings requiring model calibration and few-shot replay. To study the further usages of the NPCL's uncertainty estimation, we show its out-of-the-box readiness for novel data detection and instance-level confidence evaluation [17]. Lastly, we list the key limitations of the NPCL as an attempt to lay further solid directions for uncertainty-aware continual learning.

## 2 Related Work

**Continual Learning (CL).** Existing CL methods address catastrophic forgetting through three major approaches: (a) *Regularization-based* methods penalize changes in a model's important weights for previous tasks, such as Elastic Weight Consolidation (EWC) [30], Synaptic Intelligence (SI) [54], etc. (b) *Parameter Isolation-based* methods partition the network's parameters to specialize on individual tasks, e.g., Douillard et al. [9] learning task-specific tokens for Transformers. (c) *Replay-based* methods use an episodic memory to preserve a fraction of the past tasks' experience for preventing forgetting while learning on new tasks; e.g., experience replay (ER) [6] storing past inputs, dark experience replay (DER) [4] storing past logits, and Yan et al. [52] using a loss-aware memory for ER. Our method uses (a) via regularization of distributions, (b) via task-specific latent heads, and (c) via replay of past task inputs and distributions.

**Neural Processes (NPs).** NPs were introduced to meta-learn a distribution of a family of functions modeling the data-generating process through their deterministic [13] and/or latent summaries [14]. Attentive NPs (ANPs) [27] replaced the averaging operation in NPs with a dot-product attention [48] to enhance their expressivity. NPs/ANPs rely on a global latent that limits their ability to

model observations from multiple functions. Some works address this through local latent variables that model fine-grained correlation among a subset of the observations [50]. Recently, multi-task processes (MTPs) [26] have been studied to model multiple tasks with NPs, owing to the hierarchy of task-specific latent variables conditioned on a global latent. However, existing MTPs cannot directly handle CL problem because (a) MTPs are not designed to learn on sequential tasks and thus do not handle forgetting; (b) MTPs target the multi-task learning problem where the label for an input spans the exhaustive output space of available tasks unlike the CL setup where each input may belong to one specific and unknown task, out of multiple seen tasks.

Besides, the added complexity of variational inference has limited NP applications to mostly proof-of-concept focused regression tasks [23]. The potential of NPs for large-scale classification tasks thus remains largely under-explored except in some recent works. Wang et al. [49], for instance, leverage the predictive uncertainties of NPs to decide on pseudo labels for unlabeled data in semi-supervised classification. In our work, we use NPs to handle CL with classification tasks, reflecting the benefits of principled Bayesian learning, uncertainty estimation, and easily integrated existing ER.

## 3 Preliminaries: Neural Processes

Given the data $\{(x^t, y^t)\} = (X^t, Y^t) \sim \mathcal{D}^t$ of a task $t$, the goal is to learn the mapping $F_*^t : X^t \to Y^t$ reflecting the data-generating process. NPs [13, 14] *meta-learn* the distribution over the mapping functions from the given tasks. This is equivalent to meta-learning the distribution over the predictions $p(y_i^t | x_i^t, \mathcal{C}^t)$ for the target output $y_i^t$ belonging to a target data set $\mathcal{T}^t$, given the corresponding target input $x_i^t$ and a context set $\mathcal{C}^t$ [14, 23]. To reflect the meta-learning behavior of NPs [14], the training samples are split into the context set $|\mathcal{C}^t| = m$ and a target set $|\mathcal{T}^t| = m + n$ containing context set $\mathcal{C}$ and additional samples. NPs learn the Gaussian priors and posteriors using a neural network $F_{[\phi;\theta]}^t \approx F_*^t$ for the predictive distribution, where $\phi$ and $\theta$ parameterize an encoder $q$ and a decoder $p$, respectively. This involves deriving a global variable $z^G$ to estimate the prior $p(z^G | \mathcal{C}^t; \phi)$, and then maximizing the marginal likelihood $p(Y_{\mathcal{T}}^t | \mathcal{C}^t, X_{\mathcal{T}}^t; \theta)$:

$$p(Y_{\mathcal{T}}^t | X_{\mathcal{T}}^t, \mathcal{C}^t) = \int p(Y_{\mathcal{T}}^t | X_{\mathcal{T}}^t, z^G) p(z^G | \mathcal{C}^t) dz^G, \tag{1}$$

where $p(Y_{\mathcal{T}}^t | X_{\mathcal{T}}^t, z^G) = \prod_{i=1}^{m+n} p(y_i^t | x_i^t, z^G)$ is the generative likelihood. In CL with streaming tasks, maintaining the memorization of the task prior $p(z^G | \mathcal{C}^t; \phi)$ can help NPs avoid forgetting the *t-th* task. Our aim behind enabling NP for CL is to seek a trade-off to preserve such task priors while sharing the parameters among tasks.

## 4 Continual Learning with Neural Processes

CL considers learning from a series of different tasks arriving sequentially, *i.e.*, $\mathcal{D}^t \mid 0 \leq t \leq T - 1$. Here, $\mathcal{D}^t$ can belong to classification tasks with different classes in *class incremental* CL [8]. Let $l$ be the cross-entropy (CE) loss for classification, the CL objective for the task $t$ involves minimizing:

$$\mathcal{L}_{CE}^t = \mathbb{E}_{(x,y) \sim \mathcal{D}^t} \, l(F_{[\phi;\theta]}(x), y) \tag{2}$$

on all $[0, t]$ tasks seen sequentially. Achieving Eq. (2) is challenging in real-world CL scenarios, where the previous datasets can be unavailable due to constraints on privacy, storage, etc. Learning on the sequential data with varying distributions causes *catastrophic forgetting*. To alleviate the issue, *experience replay* (ER) is used in CL to store and periodically revisit some past experiences, e.g., samples $(x^t, y^t)$ of task $t$, in a small episodic memory $\mathcal{M}$ for replay in the future [8, 6]. In this work, we develop our method with the classical reservoir sampling-based ER [6] for a task boundary-agnostic updating of $\mathcal{M}$.

CL methods with ER solely still suffer from severe forgetting issues [6, 52, 8]; and jointly optimizing parameters on $\mathcal{D}^t$ and $\mathcal{M}$ has several drawbacks [35, 4]. Considering that a deterministic mapping $F^t$ limits capturing the randomness behind the real-world data in a stream, to utilize the meta-learning ability of NPs, we next propose extending models with Eq. (2) and Eq. (1) to arrive at our NPCL model. It allocates small subsets of parameters to learn robust per-task and global priors and uses stochastic factors to meet data-driven challenges such as deducing the right parameters for inference.

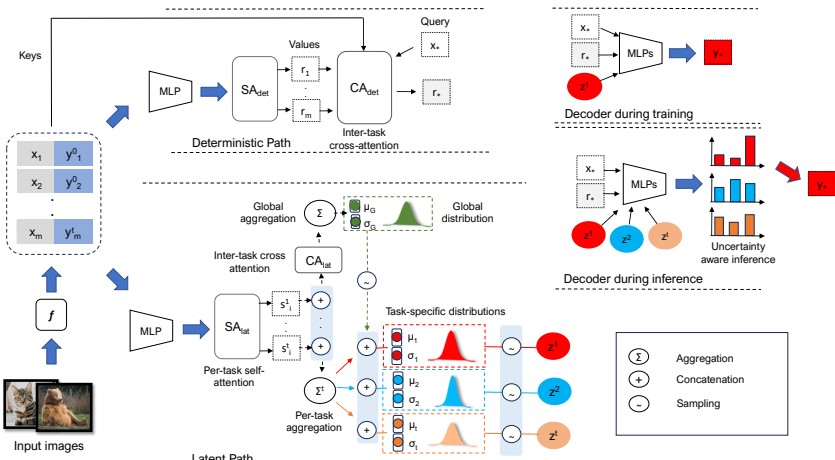

Figure 2: **Overview of the NPCL architecture**: the decoding mechanism differs during training and inference. **Red**, **Cyan**, and **Orange** denote three different tasks.

## 4.1 Neural Processes for Continual Learning

Given the task in a stream, we model the CL task based on NPs formulated in Eq. (1). In ER framework with a small memory buffer, where the context and target could be from tasks indexed by $t$, Eq. (1) can be extended to derive the joint posterior for NPs [14] as:

$$p(Y_{\mathcal{T}}^{0:t}|\mathcal{C}^{0:t}, X_{\mathcal{T}}^{0:t}) = \int p(Y_{\mathcal{T}}^{0:t}|X_{\mathcal{T}}^{0:t}, z^G)p(z^G|\mathcal{C}^{0:t})dz^G, \tag{3}$$

where $z^G$ models the joint distribution $F_*^{0:t}$ of CL tasks and is an enabler of the knowledge transfer [35] between these (see App. A.3 for ELBO). Eq. (3) poses two challenges. First, a labeled context $\mathcal{C}$ is needed for inferring predictions as all NPs, which is unprepared in CL setups by default. To overcome this, we use the memory $\mathcal{M}$ offered by the ER-based setups as *context* during inference. Second, jointly modeling $F_*^{0:t}$ ignores the dynamics of per-task stochasticities and is still prone to the bottlenecks of Eq. (2). We address the issue by introducing hierarchical modeling and redefining Eq. (3) in the following.

## 4.2 NPs with Hierarchical Task-specific Priors for CL

To learn informative task priors while tackling the forgetting issue in CL, we propose a hierarchical modeling of the NP model. We preserve the global latent $z^G$ to induce the direct knowledge transfer and add the task-specific upon the global variable to enhance the capturing of task-specific knowledge in CL. We thus extend Eq. (3) with task-specific latent variables $z^t = (z^0, .., z^t)$. As a result, our posterior is a two-step hierarchical latent variable model (Fig. 2) where the global and the per-task latent variables model the inter and intra-task correlations, respectively:

$$p(Y_{\mathcal{T}}^{0:t}|X_{\mathcal{T}}^{0:t}, \mathcal{C}^{0:t}) = \int \int \Big[ \prod_{t=0}^{T-1} p(Y_{\mathcal{T}}^{t}|X_{\mathcal{T}}^{t}, z^t)p(z^t|z^G, \mathcal{C}^t) \Big] p(z^G|\mathcal{C}^{0:t})dz^{0:t}dz^G, \tag{4}$$

where the entire context $\mathcal{C}^{0:t}$ is first encoded into $z^G$ and then conditioned on $z^G$, the task-specific context $C^t := (C^0, .., C^t)$ are encoded into their respective latent variables. We refer to Eq. (4) as NP for CL (NPCL). The hierarchical modeling enables NPCL to learn the shareable knowledge via $z^G$ and the task-specific knowledge via $z^t$ in the meta-learning fashion of NPs. Task identity is used in training to specify the task-specific latent variables. Unlike MTP [26] making predictions of all tasks for all inputs, NPCL needs to specify the corresponding output space for each test sample. We further discuss the relationship between NPCL and NP-based meta-learning in App. B.

## 4.3 The NPCL Architecture

As standard NPs [14, 27], the training samples are split into a context $\mathcal{C}$ and a target set $\mathcal{T}$ containing $\mathcal{C}$ and additional samples. Given the inputs $x_i$ from $\mathcal{C}$ or $\mathcal{T}$, we first pass these to a feature extractor

$f$. With a slight abuse of notation, we denote the features as $x_i : x_i \in \mathbb{R}^{|f|}$ and let $|f|$ denote the dimension. $x_i$ concatenated with the one-hot encoded labels, *i.e.*, $[x_i; y_i]$, is fed to the NPCL encoder with a deterministic and a latent path, and then to the decoder (Fig. 2). All the NPCL layers use multi-layer perceptrons (MLPs) projections, *i.e.*, $\text{MLP}(x) : \mathbb{R}^{|f|} \to \mathbb{R}^{|o|}$, where $|o|$ is the output feature dimension as a hyperparameter. We denote a normal distribution with a mean $\mu$ and a variance $\sigma^2$ by $\mathcal{N}(\mu, \sigma^2)$; the global and the task-specific distributions are $\mathcal{N}(\mu_G, \sigma_G^2)$ and $\mathcal{N}(\mu_t, \sigma_t^2)$.

**Latent Encoder.** The latent path comprises of the projection $\Phi_i^{\text{lat}} = \text{MLP}([x_i; y_i])$ followed by two attention operations [48]. First, per-task projections form the keys, values and queries to taskwise self-attention layers $SA_{lat}^t$ that produce order-invariant encodings $s_i^t$ over the samples of task $t$. Second, all encodings $\{s_i^{0:t}\}_{i=1}^{n+m}$ serve as the keys, values and queries to cross-attention layers $CA_{\text{lat}}^{0:t}$ that enrich their order-invariance from intra-task $s^t$ to inter-task $s^G$. $s^t$ and $s^G$ are used to derive the $N$ and $M$ Monte Carlo samples of the global $z^G$ and the task-specific latent variables $z^t$, respectively (see App. C for more details) using the reparameterization trick [29]. We set $M = 1$ to enhance the inter-task stochasticity in posterior while retaining superior computational efficiency (see App. E). For each input, we thus get $N \times (t + 1)$ latent outputs.

**Deterministic Encoder.** The deterministic path is similar to that of the ANP [27] and outputs an order-invariant representation $r_*$ for target $x_*$ (see App. C).

**Decoder.** Based on the task information, the decoder adopts separate mechanisms during training and inference. At train time, we use the available task labels to filter the $N$ true latent variables $\{z_i^t\}_{i=1}^N$, combine them with $r_*$ and $x_*$, and decode the logits $h_*$. We discuss the decoding operation in the testing phase without task ID in Sec. 4.5.

## 4.4 Learning Objectives for the NPCL

The learning of the NPCL involves variational inference alongside additional regularizations.

**Evidence Lower Bound (ELBO).** The intractability of Eq. (4) leads us to the following ELBO:

$$
\begin{aligned}
\log p_\theta(Y_{\mathcal{T}}^{0:t}|X_{\mathcal{T}}^{0:t}, \mathcal{C}) \geq \mathbb{E}_{q_\phi(z|\mathcal{T})}\Big[ &\sum_{t=0}^{T-1} \mathbb{E}_{q_\phi(z^t|z^G, \mathcal{C}^t)}[\log p_\theta(Y_{\mathcal{T}}^t|X_{\mathcal{T}}^t, z^t)] \\
&- D^t\Big(q_\phi(z^t|z^G, \mathcal{T}^t)\|q_\phi(z^t|z^G, \mathcal{C}^t)\Big)\Big] - D^G\Big(q_\phi(z^G|\mathcal{T})\|q_\phi(z^G|\mathcal{C})\Big),
\end{aligned}
\tag{5}
$$

where $p_\theta(Y_{\mathcal{T}}^t|X_{\mathcal{T}}^t, z^t)$ is approximated by the CE loss. $D^t$ and $D^G$ denote the KL divergence (KLD) between the approximate posterior and prior for the task-specific and global distributions, respectively. We derive the ELBO in App. A.1. We next propose two techniques to counter forgetting in the NPCL. Henceforth, we use $D$ to denote the Jenshen-Shannon (JS) divergence [11] between two distributions.

**Global Regularization (GR).** The training data of a CL task $t$ is dominated by the t-*th* task samples. For the NPCL, this drifts the global distribution $\mathcal{N}(\mu_G^t, \sigma_G^t)$ of past tasks towards the new task (Fig. 1). We thus regularize their global distribution using the one learned at step $t - 1$:

$$
\mathcal{L}_{\text{GR}} = D\big(\mathcal{N}(\mu_G, \sigma_G^2)_t, \mathcal{N}(\mu_G, \sigma_G^2)_{t-1}\big)
\tag{6}
$$

**Task-specific Regularization (TR).** While GR helps preserve the joint distribution of the past tasks, the hierarchy in the NPCL leaves their task-specific distributions to be still prone to forgetting (Fig. 3(a)). This can further amplify the posterior collapse [46] for past task-specific latent variables during CL training (Fig. 3(b)). To alleviate these, we regularize the learning of previous task distributions as:

$$
\mathcal{L}_{\text{TR}}^t = D\big(\mathcal{N}(\mu_t, \sigma_t^2)_t, \mathcal{N}(\mu_t, \sigma_t^2)_j\big),
\tag{7}
$$

where $j$ is the step at which the task $t$ arrived. Given the reliance of Eq. (6) and Eq. (7) on past distributions, we maintain a separate buffer, which we refer to as the distribution memory $\mathcal{M}_{\mathcal{N}}$, to store the global $\mathcal{N}(\mu_G, \sigma_G^2)$ and the task-specific distributions $\mathcal{N}(\mu_{0:t-1}, \sigma_{0:t-1}^2)$. $\mathcal{M}_{\mathcal{N}}$ is updated after each incremental training step, where we run an additional pass over the training data of task $t$ alongside replaying $\mathcal{M}$ to record the batchwise averaged global and task-specific means and variances.

**Integrated objective.** Using $\alpha$, $\beta$, $\gamma$, and $\delta$ to denote the loss weights, our total loss can be written as:

$$
\mathcal{L} = \frac{1}{|\mathcal{D}^t| + |\mathcal{M}|} \sum_{(x^t, y^t) \in \mathcal{D}^t \cup \mathcal{M}} (\mathcal{L}_{\text{CE}} + \alpha D^t + \beta D^G) + \frac{1}{|\mathcal{M}|} \sum_{(x^t, y^t) \in \mathcal{M}} \gamma \mathcal{L}_{\text{GR}} + \delta \mathcal{L}_{\text{TR}}^t,
\tag{8}
$$

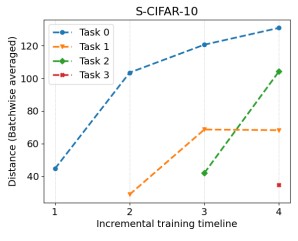

(a) Drift of past task distributions

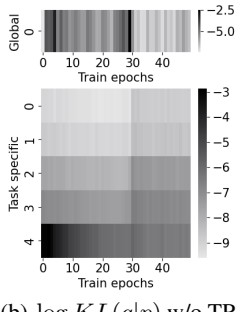

(b) $\log KL(q|p)$ w/o TR

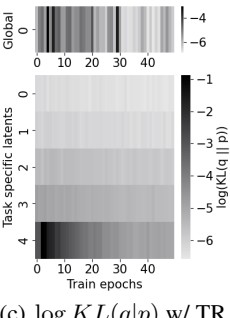

(c) $\log KL(q|p)$ w/ TR

Figure 3: **Analyses on the need for distribution regularization:** (a) shows the increasing distances between current distributions of past tasks and their original distributions (learned while the tasks were introduced). (b) and (c) show the effect of global (GR) and task regularization (TR) on the activation of the global and task-specific latent units. Low KLD corresponds to an inactive unit.

where CE, $D^t$, and $D^G$ act on the current task data $\mathcal{D}^t$ and on the buffer $\mathcal{M}$ while GR and TR act only on $\mathcal{M}$. By setting $0 < \{\alpha, \beta, \gamma, \delta\} < 1$, we resort to using the (respective) cold posteriors [55].

### 4.5 Inference with Uncertainty Awareness

NPCL's inference uses $f$ to obtain the features $x_*$ for the target test images. Although the task identification information is used to train the task-specific module, task identification of test samples is usually unavailable in general real CL tasks (except the restricted task-incremental setting [8]). Given $x_*$ from the encoder, this leaves us with $\{z_i^{0:t}\}_{i=1}^N$ possible modules and the corresponding latent variables to use and infer for obtaining the prediction. A *naive* solution is to average over $N * (t + 1)$ logits. But as the number of tasks grows, the noise from incorrect task priors would dominate the posterior. We thus propose using entropy as an uncertainty quantification metric (UQM) to filter the logits of the true task head $\psi^t$:

$$h_* = \arg\min_{j \in [1,t]} U(h_{\psi^j}), \quad U(h_\psi) = -\sum_{i \in N} \delta(i) \log(\delta(i)), \tag{9}$$

where $\delta$ is the softmax function and $U$ is the total Shannon entropy [45] over the $N$ logits per head. As we use true head $\psi_\phi^t$ during training, $\phi \in \psi^t$ produces low entropy for within distribution data. In light of Eq. (9), the NPCL can be seen as a special case of the mixture-of-expert (MoE) modeling [36, 51], where we leverage uncertainty to select the top-1 expert during inference.

## 5   Experiments

**Datasets.** We evaluate the NPCL on class and domain incremental learning (IL) settings. For class-IL, we use three public datasets: sequential CIFAR10 (S-CIFAR-10) [35], sequential CIFAR100 (S-CIFAR-100) [54], and sequential Tiny ImageNet (S-Tiny-ImageNet) [6]. For domain-IL, we use Permuted MNIST (P-MNIST) [30] and Rotated MNIST (R-MNIST) [35]. S-CIFAR-10, S-CIFAR-100, and S-Tiny-ImageNet host 10, 100, and 200 classes each with 5000, 500, and 500 training images and 1000, 100, and 50 test images per class, respectively. The number of sequential tasks for S-CIFAR-10 is 5 (2 classes per task); for S-CIFAR-100 and S-Tiny-ImageNet is 10 (10 and 20 classes per task, respectively); for P/R-MNIST is 20. P-MNIST creates tasks out of MNIST [32] by randomly permuting the pixels, and R-MNIST does it by rotating images randomly in $[0, \pi)$.

**Architectures.** For a fair comparison against other methods, we rely on the Mammoth CL benchmark [3]. Our backbone for class-IL experiments is a ResNet-18 [20] without pretraining, while for domain-IL, we rely on a fully connected (FC) network with two hidden layers [35]. The NPCL relies on Xavier initialized [15] FC layers with two 256-d hidden layers for class-IL and one 32-d layer for domain-IL setups. For class-IL, each FC layer is followed by layer normalization [1] and ReLU.

**Configuration and hyperparameters.** We train all models using SGD optimizer. The number of training epochs per task for S-Tiny-ImageNet is 100, for S-CIFAR-(10/100) is 50, and that for (P/R)-MNIST is 1. We detail further the configurations, hyperparameters, and their tuning in App. D.

| Method | S-CIFAR-10 Class-IL | | S-CIFAR-100 Class-IL | | S-Tiny-ImageNet Class-IL | | P-MNIST Domain-IL | | R-MNIST Domain-IL | |
|---|---|---|---|---|---|---|---|---|---|---|
| Joint ResNet | 92.2±0.15 | | 70.44 | | 59.99±0.19 | | 94.33±0.17 | | 95.76±0.04 | |
| Joint NP | 91.66±0.11 | | 70.58±0.24 | | 59.83±0.17 | | 95.02±0.21 | | 95.37±0.07 | |
| Joint ANP | 91.26±0.16 | | 70.77±0.21 | | 60.14±0.17 | | 95.39±0.18 | | 95.85±0.05 | |
| Joint NPCL | 92.74±0.12 | | 71.46±0.20 | | 60.18±0.22 | | 95.97±0.14 | | 96.11±0.03 | |
| Multitask NPCL | 69.15±0.09 | | 53.6±0.21 | | 35.53±0.13 | | 87.40±0.10 | | 89.21±0.02 | |
| oEWC [44] | 19.49±0.12 | | - | | 7.58±0.10 | | 75.79±2.25 | | 77.35±5.77 | |
| SI [54] | 19.48±0.17 | | - | | 6.58±0.31 | | 65.86±1.57 | | 71.91±5.83 | |
| LwF [34] | 19.61±0.05 | | - | | 8.46±0.22 | | - | | - | |
| $\mathcal{M}_{size}$ | 200 | 500 | 500 | 2000 | 200 | 500 | 200 | 500 | 200 | 500 |
| ER [43] | 44.79±1.86 | 57.74±0.27 | 22.10 | 38.58 | 8.49±0.16 | 9.99±0.29 | 72.37±0.87 | 80.6±0.86 | 85.01±1.90 | 88.91±1.44 |
| iCaRL [42] | 49.02±3.20 | 47.55±3.95 | 46.52 | 49.82 | 7.53±0.79 | 9.38±1.53 | - | - | - | - |
| FDR [2] | 30.91±2.74 | 28.71±3.23 | - | - | 8.70±0.19 | 10.54±0.21 | 74.77±0.83 | 83.18±0.53 | 85.22±3.35 | 89.67±1.63 |
| RPC [41] | - | - | 22.34 | 38.33 | - | - | - | - | - | - |
| DER [4] | 61.93±1.79 | 70.51±1.67 | 36.6 | 51.89 | 11.87±0.78 | 17.75±1.14 | 81.74±1.07 | 87.29±0.46 | 90.04±2.61 | 92.24±1.12 |
| NP [14] | 46.1±3.44 | 59.3±2.76 | 22.92 | 38.70 | 8.32±0.62 | 10.2±0.34 | 70.02±1.44 | 79.44±0.81 | 85.03±2.7 | 88.16±1.66 |
| ANP [27] | 46.67±1.23 | 58.77±0.65 | 23.2 | 39.06 | 8.81±0.93 | 9.75±0.90 | 73.55±0.66 | 80.98±0.57 | 85.70±1.39 | 89.21±0.93 |
| ST-NPCL (w/ only per-task latent) | 54.6±2.14 | 65.22±1.89 | 28.45 | 42.1 | 10.92±1.03 | 13.7±1.35 | 76.4±1.62 | 82.06±0.92 | 86.99±3.07 | 89.64±2.11 |
| Naive NPCL (w/o task head inf.) | 19.54±3.44 | 20.71±3.09 | 18.27 | 18.90 | 7.19±1.02 | 8.48±0.90 | 68.37±1.58 | 73.3±0.81 | 81.13±2.91 | 83.69±2.24 |
| NPCL (ours) | 63.78±1.70 | 71.34±1.48 | 37.43 | 46.71 | 12.44±0.59 | 15.29±1.02 | 83.11±0.90 | 86.52±0.77 | 91.48±1.79 | 92.07±1.39 |

Table 1: Classification accuracy for standard CL benchmarks across 10 runs. The best results are in red. The second best results are in blue. All runs of NP variants in the CL settings rely on ER. S-CIFAR-100 results are from Boschini et al. [3] while the rest are taken from Buzzega et al. [4].

**Baselines.** We employ several CL methods to compare the NPCL with. Regularization-based methods include oEWC [44] and SI [54]; knowledge distillation-based methods include iCaRL [42] and LwF [34]; rehearsal-based methods are ER [43], RPC [41], FDR [2], DER [4]. Among neural processes, we use the NP [14], the ANP [27], and the Single Task (ST) NPCL (see App. A.2) with only per-task latent variables. We use five non-CL benchmarks as upper bounds on the performances: Joint ResNet / NP / ANP / NPCL perform joint training of all tasks using a single task head while the multitask NPCL infers task heads in joint training using Eq. (9). Finally, the *naive* NPCL inference averages the logits of all task heads.

## 5.1 Results

Table 1 reports the average accuracy after training on all tasks. Across all settings, the NPCL boosts the performance of the ER and achieves either comparable or better results against the state-of-the-art (SOTA), *e.g.*, DER. Compared to the regularization-based oEWC and SI, the NPCL obtains a significant gain in performance. This is because the former methods calculate weight importance, which is liable to changes with new tasks. Regularizing explicitly towards the global and per-task distributions of past tasks helps the NPCL overcome this. Further, on both class and domain-IL, the NPCL stands out in the most challenging setting where the episodic memory size is the smallest. On domain-IL where the shift occurs within the domain instead of the classes, the performance of a number of methods degrade as they forget the relations among a task's classes. Preserving the tasks' distributions helps the NPCL maintain valuable information in this case. Analyzing the backward transfer (BWT) scores [39] shows that the NPCL's forgetting is competitive or lesser than the SOTA (see Table 10). Lastly, we note that the ST-NPCL with no hierarchy lags in BWT and accuracy due to limited knowledge transfer between tasks.

## 5.2 Ablation Studies

**Why uncertainty-aware inference works?** For our uncertainty-aware task head inference mechanism to be effective, a CL model must produce probabilities that align well with the ground truth labels of the test samples. We thus ablate the calibration errors for different CL baselines using the well-established Expected Calibration Error (ECE) [16] and Adaptive Calibration Error (ACE) [38] metrics. Table 2 shows that the NPCL has the least calibration error across S-CIFAR-10 ($\mathcal{M}_{size} = 200$) and S-CIFAR-100 ($\mathcal{M}_{size} = 500$). In general, the probabilistic nature of the ANP [27] and

| Method | S-CIFAR-10 | | S-CIFAR-100 | |
|---|---|---|---|---|
| Metric | ECE | ACE | ECE | ACE |
| ER [43] | 0.4553 | 0.8532 | 0.6459 | 0.9499 |
| DER [4] | 0.2991 | 0.8391 | 0.2484 | 0.9447 |
| ANP [27] | 0.34 | 0.8495 | 0.5441 | 0.9477 |
| NPCL (ours) | **0.2103** | **0.8155** | **0.1995** | **0.9421** |

Table 2: Model calibration errors averaged across 10 runs.

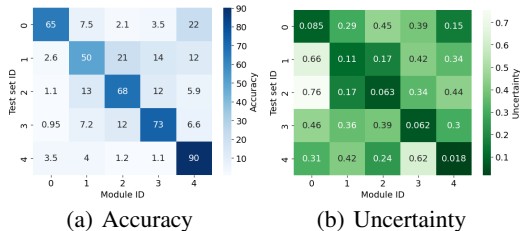

(a) Accuracy      (b) Uncertainty

Figure 4: Heatmaps depicting the taskwise averaged accuracy and uncertainty of test samples per task head on S-CIFAR-10.

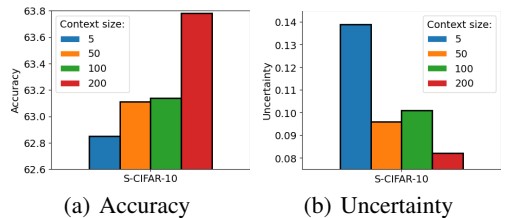

(a) Accuracy      (b) Uncertainty

Figure 5: Effect of context set size ($\mathcal{M}_{size} \in \{5, 50, 100, 200\}$) on the accuracy and uncertainty of the NPCL on S-CIFAR-10.

the NPCL benefits them in confidence calibration over the deterministic methods with comparable accuracies, *i.e.,* ER [43] and DER [4].

**Uncertainty-Accuracy trade-off.** Fig. 4 ablates the average accuracies and uncertainties of each task head predictions over the test set of each task on S-CIFAR-10 (see App. G.2 for S-CIFAR-100). First, we observe that the accuracy of predictions made by true task heads are, in general, a magnitude higher than the rest. For uncertainty, this trend is reversed. This verifies our assumption that restricting latent heads to learn only their true label distribution makes them more confident in modeling the within-task samples. Second, for recently trained tasks, the uncertainty differences between the true task heads and the rest are greater than the earlier tasks. This, in general, suggests that the extent of forgetting goes *beyond* a CL model's accuracy and to other aspects of its learning such as its predictive confidence. To support the latter claim, we probe the BWT of uncertainty and see a strong correlation with the BWT of accuracy (see Fig. 7).

**Learning objectives.** Table 3 shows the impact of distribution regularization, with the baseline being the NPCL trained with no regularization. We observe that the baseline performs worse than the ER as the NPCL layers forget more. Including TR in our objectives leads to the single-most gain over the baseline. We further study how these objectives guide the learning of the global and task-specific distributions with training (see App. G.1). We observe that the NPCL w/ TR leads to better learning of the current task as well as preserving the

| Method | S-CIFAR-10 | S-Tiny-ImageNet |
|---|---|---|
| ER [43] | 44.79 | 8.49 |
| Baseline (w/o GR or TR) | 32.24 | 7.15 |
| NPCL (w/ only GR) | 50.68 | 8.61 |
| NPCL (w/ only TR) | 57.28 | 11.36 |
| NPCL (w/ GR and TR) | **63.78** | **12.44** |

Table 3: Accuracy w/ learning objectives

past task distributions but at the cost of drifting the global distribution. The NPCL w/ GR restricts the drift of the global distribution but not for the per-task distributions. The NPCL w/ GR and TR strikes a balance in between.

**Effect of Monte Carlo (MC) samples.** We spot two combinations of the number of global $N$ and task-specific $M$ MC samples in favor of performance. Out of these, we choose the one with the superior computational efficiency (see App. E for details).

**Context size.** We study the average accuracy (Fig. 5(a)) and uncertainty (Fig. 5(b)) after training on S-CIFAR-10 with $|\mathcal{M}| = 200$, and then varying the context sizes during inference. Similar to other NPs [49, 12], we find a positive correlation between context size and performance, indicating that the NPCL utilizes useful information from diverse contexts, thereby reducing its task inference ambiguity.

**Few-shot replay settings.** A key strength of NPs remains their few-shot learning capability. To study how well the NPCL retains this trait against other CL baselines, we ablate their accuracy and ECE [16] on rehearsal memory sizes of 5 and 10 (see Table 4). We find that on both memory sizes, the NPCL outperforms ER [43] and DER [4]. For $\mathcal{M}_{size} = 5$ on S-CIFAR-100, we observe that the ER outperforms the DER in terms of accuracy. However, the latter still offers more confident predictions (characterized by a lower ECE). This implies that on few-shot CL replay settings while regularizing the predicted logits towards their old forms – as done by the DER – helps improve the predictive confidence over the ER, regularizing the task distributions towards their old forms – as done by the NPCL – remains the superior way to enhance the model's predictive confidence.

| Method | S-CIFAR-10 | | | | S-CIFAR-100 | | | |
|---|---|---|---|---|---|---|---|---|
| $\mathcal{M}_{\text{size}}$ | 5 | | 10 | | 5 | | 10 | |
| Metric | Acc. | ECE | Acc. | ECE | Acc. | ECE | Acc. | ECE |
| ER [43] | 22.11 | 0.7281 | 25.39 | 0.696 | 9.44 | 0.8003 | 9.69 | 0.8014 |
| DER [4] | 21.05 | 0.5931 | 25.2 | 0.5107 | 8.96 | 0.4593 | 10.97 | 0.542 |
| NPCL (ours) | **22.98** | **0.4709** | **26.15** | **0.441** | **10.22** | **0.39** | **12.64** | **0.4717** |

Table 4: Few-shot replay results: Accuracy (Acc.) and ECE [16] of different methods with very small buffer sizes of 5 and 10 for S-CIFAR-10 and S-CIFAR-100 averaged across 3 runs.

**Storage efficiency.** For each task, the NPCL stores two new vectors – task-specific mean and variance, and replaces the global mean and variance with the current global ones. The NPCL storage thus scales constantly in the size $|\mathcal{M}|$ of the memory. This offers a strong edge on storage efficiency when compared to DER [4] scaling quadratically, *i.e.*, $|\mathcal{M}| \times N_C$ where $N_C$ is the total number of classes. For instance, on S-Tiny-ImageNet with $|\mathcal{M}| = 500, N_C = 200$, the NPCL's cumulative storage amounts to a (flattened) vector of size 6132 ($256 \times 10 \times 2$ for 256-d means and variances of 10 tasks plus $256 \times 2$ for 256-d global mean and variance plus 500 for 1-d task labels) while that of DER amounts to 100,000 ($200 \times 500$ for logits of 500 memory samples), *i.e.,* a **93.868%** relative storage efficiency. We report the storage efficiency of the NPCL over DER across all settings in App. G.3.

## 5.3 Applications of Uncertainty Quantification

The probabilistic nature of the NPCL offers it an edge in leveraging data-driven UQMs. To further study the usage of predictive uncertainties, we conduct two experiments with a trained NPCL model.

**Novel data identification.** Novel data identification seeks to distinguish out-of-distribution data ($\mathcal{D}_{\text{OOD}}$) from in-domain data ($\mathcal{D}_{\text{ID}}$). Forgetting makes CL models struggle further on the task [19]. The probabilistic sampling in the NPCL opens the door for leveraging its predictive variances – which are more reliable estimates of aleatoric uncertainty than pointwise predictions [22]. For the $N$ predicted logits, we thus compute the variances over their softmax scores, $\sigma^2(\delta(h_*))$, and over their uncertainty scores, $\sigma^2(U(h_*))$. Table 5 evaluates these metrics for ID (S-CIFAR-10) and OOD (first 10 classes of S-CIFAR-100) data after each task. We observe that the variance scores of either metrics on $\mathcal{D}_{\text{ID}}$ are up to a magnitude lower than those on $\mathcal{D}_{\text{OOD}}$. We further observe an overall decrease in the variances with the arrival of further incremental tasks. This could be attributed to the generalization of more low-level features in the novel data as in-domain [18, 16]. We detail further novel data identification experiments in App. G.5.

| Incremental step | $\mathcal{D}_{\text{ID}} = $ CIFAR-10, $\mathcal{D}_{\text{OOD}} = $ CIFAR-100 | | | |
|---|---|---|---|---|
| | $\mathcal{D}_{\text{ID}}$ $(\delta)$ | $\mathcal{D}_{\text{OOD}}$ $(\delta)$ | $\mathcal{D}_{\text{ID}}$ (H) | $\mathcal{D}_{\text{OOD}}$ (H) |
| 1 | $1e^{-6}$ | $1e^{-5}$ | $9.3e^{-6}$ | $8.4e^{-5}$ |
| 2 | $2.6e^{-6}$ | $1.4e^{-5}$ | $6.3e^{-5}$ | $2.2e^{-4}$ |
| 3 | $2.3e^{-6}$ | $6.2e^{-6}$ | $6.7e^{-5}$ | $2.1e^{-4}$ |
| 4 | $8.1e^{-7}$ | $4.8e^{-6}$ | $4.6e^{-5}$ | $2.2e^{-4}$ |
| 5 | $7.1e^{-7}$ | $1.7e^{-6}$ | $4.6e^{-5}$ | $1.1e^{-4}$ |

Table 5: Average variances over softmax $(\delta)$ and entropy $(H)$ scores on in-domain and out-of-domain test sets using $N = 50$ ancestral samples.

**Instance-level model confidence evaluation.** The confidence evaluation framework of Han et al. [17] provides finer granularity for assessing the predictive confidence of classification models (see App. G.6 for more details and normality test). Table 6 shows the results of one run of the framework after training on S-CIFAR-10. Here, we use the task identity to select the latent head per class. We observe the mean prediction interval width (PIW) of the true class label among the correct predictions to be narrower than that of the incorrect predictions, implying that the NPCL's variations of predicted class labels are smaller when the predictions are correct. We also notice a higher accuracy among the test instances rejected by the *t*-test than those not rejected.

| Class | Accuracy | PIW | | Accuracy by $t$-test status | |
|---|---|---|---|---|---|
| | | Correct | Incorrect | Rejected | Not Rejected |
| 1 | 82.30 | 74.17 | 102.21 | 83.37 | 50.00 |
| 2 | 94.00 | 62.90 | 79.86 | 94.07 | 80.00 |
| 3 | 74.00 | 54.92 | 68.48 | 74.14 | 64.29 |
| 4 | 71.50 | 65.42 | 74.32 | 72.06 | 25.00 |
| 5 | 84.80 | 92.93 | 106.90 | 85.37 | 22.22 |
| 6 | 76.50 | 75.22 | 103.58 | 76.58 | 60.00 |
| 7 | 94.20 | 104.9 | 129.56 | 94.39 | 3.00 |
| 8 | 90.50 | 81.10 | 127.06 | 91.12 | 22.22 |
| 9 | 96.90 | 72.81 | 110.86 | 97.00 | 66.67 |
| 10 | 96.30 | 80.60 | 109.56 | 96.48 | 60.00 |

Table 6: PIW (multiplied by 100) and $t-$test results for the first three classes of S-CIFAR-10 inferred from their respective task heads.

# 6 Limitations

We list the key limitations of the NPCL to facilitate future research directions.

**Incompetence of dot-product attention.** Similar to the ANP [27], the NPCL employs the permutation-invariant scaled-dot product attention [48] to weigh the relevant context and target embeddings. Visualizing the attention weights computed by the cross-attention layers of the deterministic path shows us that the top attended context for the target queries often contain points belonging to other CL tasks (Fig. 11(a)). This *limits* the performance sensitivity of the NPCL with respect to the increase in context thus resulting in a lag of accuracy behind SOTA on CL setups with larger episodic memory sizes (see Table 1). To further verify the relevance of the attended context, we visualize the self-attention weights of all context points. Fig. 11(b) shows that the lowest or the maximum values in the context dataset have larger weights. Such an observation is in line with existing works pointing that the scaled-dot product attention can derive irrelevant set encodings of the context points and can thus lag at exploiting the context embeddings properly [28].

**Computational overhead.** Table 7 compares the number of parameters of the NPCL with ER / DER [3] where the latter rely solely on the ResNet-18 backbone as they do not exploit parameter isolation for task heads. Overall, the percentage increase in parameter number is $57.6\%$ for S-CIFAR-10, $46.57\%$ for S-CIFAR-100 and S-Tiny-ImageNet, and $55.25\%$ for P/R-MNIST.

| Method / Dataset | S-CIFAR-10 | S-CIFAR-100 | S-Tiny-ImageNet | P/R-MNIST |
|---|---|---|---|---|
| ER / DER [3] | 11,173,962 | 11,220,132 | 11,220,132 | 89,610 |
| NPCL | 19,397,706 | 24,091,556 | 24,091,556 | 162,166 |

Table 7: Comparison of the total number of parameters for ER / DER against the NPCL.

**Inference time complexity.** The reliance on self-attention means that the inference time complexity of the NPCL is $\mathcal{O}(n * m)$, where $n$ is the number of context points (sampled from the episodic memory) and $m$ is the number of target points (the number of test samples). Due to this, the runtime for inference scales polynomially with the number of context points (sampled from the buffer). Table 6 reports the runtime of the NPCL on S-CIFAR-10 and S-CIFAR-100 settings by varying the context sizes. For reference, the first row reports the runtime of ER / DER whose inference complexity is $\mathcal{O}(1)$ in the memory buffer size.

| Method | S-CIFAR-10 |
|---|---|
| ER / DER | 3.72s |
| NPCL, $|\mathcal{M}| = 200$ | 19.58s |
| NPCL, $|\mathcal{M}| = 500$ | 31.25s |
| NPCL, $|\mathcal{M}| = 1000$ | 47.99s |
| NPCL, $|\mathcal{M}| = 2000$ | 84.86s |

Table 8: Inference time with varying context sizes

**Incompatibility with logits-based replay.** The NPCL is incompatible with logits-based replay because of the stochasticity in the posterior induced by Monte Carlo sampling. Overcoming this could help boost the performance of the NPCL further over SOTA like DER [4] and DER++ [3].

# 7 Conclusion

In this paper, we propose Neural Processes for Continual Learning (NPCL), a hierarchical latent variable setup designed to jointly model the task-agnostic and task-specific data-generating functions in continual learning. We study the potential forgetting aspects in the NPCL and propose to regularize the previously learned distributions at a global and a per-task granularity. We demonstrate that using entropy as an uncertainty quantification metric helps the NPCL infer correct task heads and boost the performance of baseline experience replay to even surpass state-of-the-art deterministic models on several CL settings. Our robust ablations show the efficacy of the NPCL for model calibration measurement and few-shot replay in CL. We further study out-of-the-box applications of the uncertainty estimation capabilities of the NPCL for novel data identification and instance-level confidence evaluation. We conclude our ablations by listing the key limitations of the NPCL, which we hope could lay solid directions for further research on uncertainty-aware continual learning.

## Acknowledgment

This work was partially supported by an ARC DECRA Fellowship DE230101591 awarded to Dong Gong. We acknowledge the reviewers for their valuable feedback.

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

# A   Theory

## A.1   ELBO derivation for the NPCL

Borrowing the conventions from Sec. 3, for an incremental task $0 \leq t \leq T$, we assume the context $\mathcal{C}$ and targets $\mathcal{T}$ to comprise of samples from all $t$ seen classes. Accordingly, we define these as $\mathcal{C} = (X_{\mathcal{C}}^{0:t}, Y_{\mathcal{C}}^{0:t})$ and $\mathcal{T} = (X_{\mathcal{T}}^{0:t}, Y_{\mathcal{T}}^{0:t})$, respectively. To enforce the prior that both $\mathcal{C}$ and $\mathcal{T}$ follow the same distribution, we assume $\mathcal{C}^t \subset \mathcal{T}^t$, and therefore, $\mathcal{C} \subset \mathcal{T}$. In order to derive predictions $Y_{\mathcal{T}}^{0:t}$ on $X_{\mathcal{T}}^{0:t}$, the NPCL relies on the context $\mathcal{C}$ to build conditional priors $p_\theta(z^G | \mathcal{C})$ and $p_\theta(z^t | z^G, \mathcal{C}^t)$, where $p_\theta$ is the decoder. The decoder's objective thus boils down to maximizing the log-likelihood of the observations, *i.e.,* the evidence $\log p_\theta(Y_{\mathcal{T}}^{0:t} | X_{\mathcal{T}}^{0:t}, \mathcal{C})$. In the following, we derive the evidence lower bound (ELBO):

$$\log p_\theta(Y_{\mathcal{T}}^{0:t} | X_{\mathcal{T}}^{0:t}, \mathcal{C}) \qquad \text{(Log-likelihood of evidence)} \quad (10\text{a})$$

$$= \log p_\theta(Y_{\mathcal{T}}^{0:t} | X_{\mathcal{T}}^{0:t}, \mathcal{C}) \int p_\theta(z^G | X_{\mathcal{T}}^{0:t}, Y_{\mathcal{T}}^{0:t}, \mathcal{C}) dz^G \qquad \left( \because \int p_\theta(z^G | \mathcal{T}, \mathcal{C}) dz^G = 1 \right) \quad (10\text{b})$$

$$= \int p_\theta(z^G | X_{\mathcal{T}}^{0:t}, Y_{\mathcal{T}}^{0:t}, \mathcal{C})(\log p_\theta(Y_{\mathcal{T}}^{0:t} | X_{\mathcal{T}}^{0:t}, \mathcal{C})) dz^G \qquad \text{(Integrate over the log-likelihood)} \quad (10\text{c})$$

$$= \mathbb{E}_{q_\phi(z^G | \mathcal{T})}[\log p_\theta(Y_{\mathcal{T}}^{0:t} | X_{\mathcal{T}}^{0:t}, \mathcal{C})] \qquad \text{(By definition)} \quad (10\text{d})$$

$$= \mathbb{E}_{q_\phi(z^G | \mathcal{T})}\left[ \log \frac{p_\theta(Y_{\mathcal{T}}^{0:t}, z^G | X_{\mathcal{T}}^{0:t}, \mathcal{C})}{p_\theta(z^G | X_{\mathcal{T}}^{0:t}, Y_{\mathcal{T}}^{0:t}, \mathcal{C})} \right] \qquad \text{(Re-introduce } z^G \text{ by Chain rule)} \quad (10\text{e})$$

$$= \mathbb{E}_{q_\phi(z^G | \mathcal{T})}\left[ \log \frac{p_\theta(Y_{\mathcal{T}}^{0:t} | X_{\mathcal{T}}^{0:t}, \mathcal{C}, z^G) p_\theta(z^G | X_{\mathcal{T}}^{0:t}, \mathcal{C})}{p_\theta(z^G | \mathcal{T})} \right] \qquad \text{(Chain rule of probability; } \mathcal{C} \subset \mathcal{T}) \quad (10\text{f})$$

$$= \mathbb{E}_{q_\phi(z^G | \mathcal{T})}\left[ \log \frac{p_\theta(Y_{\mathcal{T}}^{0:t} | X_{\mathcal{T}}^{0:t}, \mathcal{C}, z^G) p_\theta(z^G | X_{\mathcal{T}}^{0:t}, \mathcal{C}) q_\phi(z^G | \mathcal{T})}{p_\theta(z^G | \mathcal{T}) q_\phi(z^G | \mathcal{T})} \right] \qquad \text{(Equivalent fraction)} \quad (10\text{g})$$

$$= \mathbb{E}_{q_\phi(z^G | \mathcal{T})}\left[ \log p_\theta(Y_{\mathcal{T}}^{0:t} | X_{\mathcal{T}}^{0:t}, \mathcal{C}, z^G) \right]$$
$$\quad + \mathbb{E}_{q_\phi(z^G | \mathcal{T})}\left[ \log \frac{p_\theta(z^G | \mathcal{C})}{q_\phi(z^G | \mathcal{T})} \right] + \mathbb{E}_{q_\phi(z^G | \mathcal{T})}\left[ \log \frac{q_\phi(z^G | \mathcal{T})}{p_\theta(z^G | \mathcal{T})} \right] \qquad \text{(Split the expectation)} \quad (10\text{h})$$

$$= \mathbb{E}_{q_\phi(z^G | \mathcal{T})}\left[ \log p_\theta(Y_{\mathcal{T}}^{0:t} | X_{\mathcal{T}}^{0:t}, \mathcal{C}, z^G) \right]$$
$$\quad - D_{\text{KL}}\big(q_\phi(z^G | \mathcal{T}) \| p_\theta(z^G | \mathcal{C})\big) + D_{\text{KL}}\big(q_\phi(z^G | \mathcal{T}) \| p_\theta(z^G | \mathcal{T})\big) \qquad \text{(By definition of KL divergence)} \quad (10\text{i})$$

$$\geq \mathbb{E}_{q_\phi(z^G | \mathcal{T})}\left[ \log p_\theta(Y_{\mathcal{T}}^{0:t} | X_{\mathcal{T}}^{0:t}, \mathcal{C}, z^G) \right] - D_{\text{KL}}\big(q_\phi(z^G | \mathcal{T}) \| p_\theta(z^G | \mathcal{C})\big), \qquad (\because \text{ KL divergence} \geq 0) \quad (10\text{j})$$

where the evidence is equal to the sum of the reconstruction likelihood $\mathbb{E}_{q_\phi(z^G | \mathcal{T})}\left[ \log p_\theta(Y_{\mathcal{T}}^{0:t} | X_{\mathcal{T}}^{0:t}, \mathcal{C}, z^G) \right]$ of the decoder and the KL divergence between the true posterior $p_\theta(z^G | \mathcal{T})$ and the approximate posterior $q_\phi(z^G | \mathcal{T})$ learned using the variational distribution, minus the prior matching term $D_{\text{KL}}\big(q_\phi(z^G | \mathcal{T}) \| p_\theta(z^G | \mathcal{C})\big)$. In particular, the NPCL learns two approximate distributions $q_\phi(z^G | \mathcal{T})$ and $q_\phi(z^t | z^G, \mathcal{T}^t)$, that seek to estimate the global posterior $p_\theta(z^G | \mathcal{T})$ and the task-specific posterior $p_\theta(z^t | z^G, \mathcal{T}^t)$. To realize the latter posterior, we introduce the hierarchy of task-specific latent variables $z^{0:t}$. This allows us to expand and derive a lower bound to the reconstruction likelihood as:

$$\log p_\theta(Y_\mathcal{T}^{0:t}|X_\mathcal{T}^{0:t}, \mathcal{C}, z^G) \qquad \text{(Reconstruction term)}$$
$$\text{(11a)}$$

$$= \mathbb{E}_{\prod_0^t q_\phi(z^t|z^G, \mathcal{T}^t)}\left[\log \frac{p_\theta(Y_\mathcal{T}^{0:t}, z^{0:t}|X_\mathcal{T}^{0:t}, \mathcal{C}, z^G)}{p_\theta(z^{0:t}|X_\mathcal{T}^{0:t}, Y_\mathcal{T}^{0:t}, \mathcal{C}, z^G)}\right] \qquad \text{(Introduce one-level latent hierarchy)}$$
$$\text{(11b)}$$

$$= \mathbb{E}_{\prod_0^t q_\phi(z^t|z^G, \mathcal{T}^t)}\left[\log \int_0^t \frac{p_\theta(Y_\mathcal{T}^t, z^t|X^t, \mathcal{C}^t, z^G)}{p_\theta(z^t|X_\mathcal{T}^t, Y_\mathcal{T}^t, \mathcal{C}^t, z^G)}\right] \qquad \text{(Integrate over individual tasks)}$$
$$\text{(11c)}$$

$$= \int_0^t \mathbb{E}_{q_\phi(z^t|z^G, \mathcal{T}^t)}\left[\log \frac{p_\theta(Y_\mathcal{T}^t|X_\mathcal{T}^t, \mathcal{C}^t, z^G, z^t)p_\theta(z^t|X_\mathcal{T}^t, \mathcal{C}^t, z^G)}{p_\theta(z^t|\mathcal{T}^t, z^G)}\right] \qquad \text{(Chain rule of probability; } \mathcal{C} \subset \mathcal{T})$$
$$\text{(11d)}$$

$$= \int_0^t \mathbb{E}_{q_\phi(z^t|z^G, \mathcal{T}^t)}\left[\log \frac{p_\theta(Y_\mathcal{T}^t|X_\mathcal{T}^t, z^t)p_\theta(z^t|\mathcal{C}^t, z^G)q_\phi(z^t|z^G, \mathcal{T}^t)}{p_\theta(z^t|\mathcal{T}^t, z^G)q_\phi(z^t|z^G, \mathcal{T}^t)}\right] \qquad \text{(Equivalent fraction)}$$
$$\text{(11e)}$$

$$= \int_0^t \mathbb{E}_{q_\phi(z^t|z^G, \mathcal{T}^t)}\left[\log p_\theta(Y_\mathcal{T}^t|X_\mathcal{T}^t, z^t)\right]$$
$$- D_{\text{KL}}\big(q_\phi(z^t|z^G, \mathcal{T}^t)\|p_\theta(z^t|z^G, \mathcal{C}^t)\big) + D_{\text{KL}}\big(q_\phi(z^t|z^G, \mathcal{T}^t)\|p_\theta(z^t|z^G, \mathcal{T}^t)\big) \qquad \text{(By definition of KL divergence)}$$
$$\text{(11f)}$$

$$\geq \int_0^t \mathbb{E}_{q_\phi(z^t|z^G, \mathcal{T}^t)}\left[\log p_\theta(Y_\mathcal{T}^t|X_\mathcal{T}^t, z^t)\right] - D_{\text{KL}}\big(q_\phi(z^t|z^G, \mathcal{T}^t)\|p_\theta(z^t|z^G, \mathcal{C}^t)\big) \qquad (\because \text{ KL divergence} \geq 0),$$
$$\text{(11g)}$$

Plugging Eq. (11g) into Eq. (10j), we get the final ELBO:

$$\log p_\theta(Y_\mathcal{T}^{0:t}|X_\mathcal{T}^{0:t}, \mathcal{C}) \qquad \text{(12a)}$$

$$\geq \mathbb{E}_{q_\phi(z^G|\mathcal{T})}\left[\int_0^t \mathbb{E}_{q_\phi(z^t|z^G, \mathcal{T}^t)}\left[\log p_\theta(Y_\mathcal{T}^t|X_\mathcal{T}^t, z^t)\right] - D_{\text{KL}}\big(q_\phi(z^t|z^G, \mathcal{T}^t)\|p_\theta(z^t|z^G, \mathcal{C}^t)\big)\right]$$
$$- D_{\text{KL}}\big(q_\phi(z^G|\mathcal{T})\|p_\theta(z^G|\mathcal{C})\big) \qquad \text{(By substitution)}$$
$$\text{(12b)}$$

$$= \mathbb{E}_{q_\phi(z^G|\mathcal{T})}\left[\int_0^t \mathbb{E}_{q_\phi(z^t|z^G, \mathcal{T}^t)}\left[\log p_\theta(Y_\mathcal{T}^t|X_\mathcal{T}^t, z^t)\right] - D_{\text{KL}}\big(q_\phi(z^t|z^G, \mathcal{T}^t)\|q_\phi(z^t|z^G, \mathcal{C}^t)\big)\right]$$
$$- D_{\text{KL}}\big(q_\phi(z^G|\mathcal{T})\|q_\phi(z^G|\mathcal{C})\big), \qquad \text{(Final ELBO)}$$
$$\text{(12c)}$$

where the decoder $p_\theta$ serves as the conditional prior network and is replaced by the encoder $q_\phi$ serving as the surrogate posterior network. $q_\phi$ can be seen to be producing two intermediate bottleneck distributions: (a) $q_\phi(z^G|\mathcal{T})$ transforms inputs into a distribution over global latent variables, (b) conditioned on the global latent variables, $q_\phi(z^t|z^G, \mathcal{T}^t)$ gathers the t-*th* task inputs and learns another distribution over the task-specific latent variables. The task-specific latent variables and their corresponding input covariates $X^t$ are then used by the deterministic decoder $p_\theta$ to decode their corresponding logit $h_*$. It is indeed this dependency of $p_\theta$ on the task identifier $t$ that makes inference a challenging task in real-world CL settings.

### A.2 Single Task NPCL and its ELBO

Single-Task (ST) NPCL preserves all but the inter-task cross attention $CA_{\text{lat}}^{0:t}$ and the global distribution encoder $\psi^G$ layers from the architecture of the NPCL (Sec. 4.3). The task-specific latent variables $z^t$ are thus derived as:

$$\{z_i^t\}_{i=1}^M \sim \mathcal{N}(\psi_\mu^t(s_i^t), \psi_\sigma^t(s_i^t)) = \mathcal{N}(\mu_t, \sigma_t^2), \forall t \in T, \qquad \text{(13)}$$

where $s_i^t$ and $\psi^t$ carry the same meaning as in Eq. (16). For a fair comparison in Table 1, we fix $M$ to be the same as the total number of global ancestral samples $N$ in the NPCL. The corresponding

ELBO amounts to:

$$\log p_\theta(Y_\mathcal{T}^{0:t}|X_\mathcal{T}^{0:t}, \mathcal{C}) \tag{14a}$$

$$\geq \int_0^t \mathbb{E}_{q_\phi(z^t|\mathcal{T}^t)}\Big[\log p_\theta(Y_\mathcal{T}^t|X_\mathcal{T}^t, z^t)\Big] - D_{\mathrm{KL}}\big(q_\phi(z^t|\mathcal{T}^t)\|p_\theta(z^t|\mathcal{C}^t)\big) \quad \text{(Dropping } z^G \text{ from Eq. (12c))} \tag{14b}$$

### A.3 ELBO for the NP [14] and the ANP [27]

The NP [14] and the ANP [27] employ a single latent variable $z^G$ to model the global correlation of all tasks. In particular, compared to Sec. 4.3, the task-specific self-attention layer $SA_{\mathrm{lat}}^t$ and the task-specific distribution encoder $\psi^t$ is no longer required. While this enables knowledge sharing among tasks, NPs and ANPs are limited in modeling finer intra-task stochastic factors. The ELBO can be given as:

$$\log p_\theta(Y_\mathcal{T}^{0:t}|X_\mathcal{T}, \mathcal{C}) \tag{15a}$$

$$\geq \mathbb{E}_{q_\phi(z^G|\mathcal{T})}\Big[\log p_\theta(Y_\mathcal{T}^{0:t}|X_\mathcal{T}, z^G)\Big] - D_{\mathrm{KL}}\big(q_\phi(z^G|\mathcal{T})\|p_\theta(z^G|\mathcal{C})\big) \quad \text{(Dropping } z^{0:t} \text{ from Eq. (12c))} \tag{15b}$$

where $z^G$ is derived in a way similar to Eq. (16), and the inputs $X_\mathcal{T}$ and $\mathcal{C}$ belong to $[0,t]$ tasks without relying on the task labels for being encoded.

## B NPs for Meta-Learning (ML) vs Continual Learning (CL)

**Resemblance.** For both ML and CL settings, we have multiple tasks and would like to learn an NP with flexible conditioning that generates task-specific functions. Specifically, for each task $t$, we have $Y^t = F_{[\phi;\theta]}^t(X^t; z^t)$ where the latent variable $z^t$ is conditioned on the task-specific context $\mathcal{C}^t$. As $z^t$ is task-specific, this, in turn, makes $F^t$ task-specific.

**Differences.** There are two major differences between the conventional NP for ML and our proposed NP for CL (NPCL):

1. **Architectural difference.** In standard NPs, $z^t$ is conditioned only on the task-specific context $\mathcal{C}^t$ while in the NPCL, $z^t$ is conditioned on $[\mathcal{C}^t; z^G]$ where $z^G$ is the global latent derived from the global context. The added conditioning of global latent reflects the need for cross-task knowledge transfer in CL and thus endows a two-level hierarchy into our model.

2. **Functional difference.** The functional difference between the ML and CL (inference) settings call for another adaptation in the NPCL. Namely, ML aims to learn NPs that learn $F_{[\phi;\theta]}$ that can generalize to new tasks. That is why NPs are tested on a new task $t_*$ given its context $\mathcal{C}_*$. On the other hand, in CL, we wish to learn an $F$ that can perform well on all the seen tasks as there is *no new task* during inference. Given the absence of a labeled context during inference, our test-time context samples are thus a subset of the training data from the tasks seen so far.

In a nutshell, CL calls for learning a general function that is task-specific but also performs well on all the seen tasks subject to a limited rehearsal memory. To achieve this, we leverage an NP's ability of generating task-specific functions with flexible conditioning. But instead of using the conventional NPs for ML, we propose a hierarchical model to introduce more inter-task knowledge sharing, thus tailored for the CL problem.

## C Further on the NPCL Architecture

In the following, we denote multi-head dot product self-attention [48] by $SA(K, V, Q)$ where K, V, and Q are the keys, values and queries, respectively. The equivalent notation for cross-attention is $CA(K, V, Q)$.

**Latent Encoder.** The latent path learns the functional prior and posterior from the context and the target sets, respectively. Each label-concatenated input is projected as $\Phi_i^{\mathrm{lat}} = \mathrm{MLP}([x_i; y_i])$; then

subjected to two attention operations. First, per-task projections form the keys, values, and queries to taskwise self-attention layers $SA_{\text{lat}}^t(\Phi_i^{\text{lat}}, \Phi_i^{\text{lat}}, \Phi_i^{\text{lat}}) : \Phi_i^{\text{lat}} \rightarrow s_i^t$ that produce order-invariant encodings $s_i^t$ over the task $t$. Second, all encodings $\{s_i^{0:t}\}_{i=1}^{n+m}$ serve as the keys, values, and queries to the cross-attention layers $CA_{\text{lat}}^{0:t}(s_i^t, s_i^t, s_i^t) : s_i^t \rightarrow s_i^G$ that enrich their order-invariance from intra-task $s^t$ to inter-task $s^G$. $s^t$ and $s^G$ are then used to derive the global $z^G$ and the task-specific latent variables $z^t$.

Such globally attended inputs are passed in parallel to two MLP layers constituting the global distribution encoder $\psi^G$ whose outputs together parameterize the global distribution $\mathcal{N}(\mu_G, \sigma_G^2)$ over the input set, $i.e.$, $\psi^G(s^G) : \{s_i^G\}_{i=1}^{n+m} \rightarrow (\mu_G, \sigma_G^2)$. Samples $\{z_i^G\}_{i=1}^N$ drawn from this distribution are proxies for the variables capturing the global correlation over all tasks in the input set. It is indeed this sampling step that induces the stochasticity into the learned posteriors of the NPCL.

To model finer task-specific distribution for task $t$ conditioned on the global distribution, we retain the task-specific self-attended representations $s_i^t$ and concatenate these with the global latent variables $\{z_i^G\}_{i=1}^N$ to produce $N$ distinct encodings per input point. These encodings are then passed through the t-$th$ task distribution encoder $\psi^t$ that again constitutes a mean and a variance MLP head and produces outputs that parameterize the t-$th$ task distribution $\mathcal{N}(\mu_t, \sigma_t^2)$, $i.e.$, $\psi^t(s^t) : \{s_i^t\}_{i=1}^{n+m} \rightarrow (\mu_t, \sigma_t^2)$. Samples $\{z_j^T\}_{i=j}^M$ drawn from each such distribution thus capture the per-task stochastic factors. To limit the randomness in the learned prior/posterior, we use $M = 1$. The latent encoder thus outputs a subtotal of $N \times (t + 1)$ encodings per input point.

Put together, the global and task-specific latent variables can be derived as:

$$
\begin{aligned}
\{z_i^G\}_{i=1}^N &\sim \mathcal{N}(\psi_\mu^G(s^G), \psi_\sigma^G(s^G)) = \mathcal{N}(\mu_G, \sigma_G^2), \\
\{z_i^t\}_{i=1}^M &\sim \mathcal{N}(\psi_\mu^t(s_i^t, z_i^G), \psi_\sigma^t(s_i^t, z_i^G)) = \mathcal{N}(\mu_t, \sigma_t^2), \forall t \in T,
\end{aligned}
\tag{16}
$$

where $\psi^G$ and $\psi^t$ are the global and per-task distribution encoders, respectively.

**Deterministic Encoder.** The deterministic path is similar to that of an ANP [27] where the context projections $\Phi_i^{\text{det}} = \text{MLP}([x_i; y_i])$ form the keys, queries and values for a self-attention operation, $SA_{\text{det}}(\Phi_i^{\text{det}}, \Phi_i^{\text{det}}, \Phi_i^{\text{det}}) : \Phi_i^{\text{det}} \rightarrow r_i$. The resulting order-invariant context representations $\{r_i\}_{i=1}^m$ are fed as values to a subsequent target-to-context cross-attention operation $CA_{\text{det}}$. The keys $x_i$ and queries $x_*$ for $CA_{\text{det}}$ come from the context $x_i \in X_{\mathcal{C}}$ and target $x_* \in X_{\mathcal{T}}$ covariates, respectively, $i.e.$, $CA_{\text{det}}(x_i, s_C, x_*) : x_* \rightarrow r_*$ where $r_*$ is invariant to the order of context.

**Decoder.** Different from other NP variants, the NPCL decoder adopts separate decoding mechanisms during training and inference. At train time, we use the available task identity to filter the true $N$ out of $N * (t + 1)$ latent path outputs to be processed by the decoder. After this, the decoder concatenates a target input $x_*^t \in X_{\mathcal{T}}^t$ with its $N$ $true$ task-specific latent variables $\{z_i^t\}_{i=1}^N$ obtained from the latent path and its order-invariant feature $r_*$ obtained from the deterministic path thus resulting in $N$ distinct inputs. For $N > 1$ samples of $z_t$, we first make $N$ copies of $x_*$ and $r_*$ each, and then concatenate these with each $z^t$. $p_\theta$ thus performs the projection $p_\theta([x_*; r_*; \{z_i^t\}_{i=1}^N]) : x \rightarrow h_*$ where $x \in \mathbb{R}^{f+2*o}$ and $h_*$ are the logits of an MLP classifier for the target label $y_*$. We detail the inference-time decoding in Sec. 4.5.

# D   Experiments and Reproducibility

**Configuration.**   For a fair comparison with the benchmarks of Buzzega et al. [4], we fix the batch sizes for new task's samples and for replay samples to 32 each for the class-IL datasets and to 128 each for the domain-IL datasets. Both the context and target datasets use the same set of augmentations. For S-CIFAR-10, S-CIFAR-100, and S-Tiny-ImageNet, we apply random crops and horizontal flips to both stream and buffer examples following Buzzega et al. [4] and Boschini et al. [3]. For each setting of memory size on each dataset, the NPCL adopts the same learning rate (LR) as reported in Buzzega et al. [4] and Boschini et al. [3]. However, the NPCL training additionally relies on linearly increasing the learning rate (LR) over a period of 4000 iterations for class-IL and 40 iterations for domain-IL settings. We further apply gradient clipping [40] on L2-norm of the NPCL parameters with a cap of 10000.

**Hyperparameter tuning.** We arrive at the best hyperparameter settings for each of our datasets through grid search over a validation set made of 10% of the training set on each dataset. The search range for number of samples $N$ from the global distribution $\mathcal{N}(\mu_G, \sigma_G^2)$ is $[2, 5, 10, 20, 50, 100]$. Out of these, we found $N = 50$ during training and $N = 10$ during evaluation to perform better in general across all settings.

Similarly, we conducted a grid search over the batch size of the context set $\mathcal{C}$ over the range $[1/16, 1/8, 1/4, 1/2, 1, 1.25]$ of the original (target) batch sizes for each of the dataset. In general, we found that fixing the context batch size to $1/8$ of the target batch size performed better across all datasets. Such context batches are sampled from a context dataset $\mathcal{D}_{\mathcal{C}}^t$ for each task $t$. $\mathcal{D}_{\mathcal{C}}^t$ is itself created by randomly selecting a subset of the training samples for each class at the beginning of each incremental training task. To decide on the size of the subset for each class, we ran a grid search over the range $[50, 100, 150, 200]$ samples per class and found that incorporating 100 random samples per class into $\mathcal{D}_{\mathcal{C}}^t$ performed well across all datasets.

Finally, to decide on the loss weights $\alpha$, $\beta$, $\gamma$ and $\delta$ for $D^t$, $D^G$, $\mathcal{L}_{\text{GR}}$, and $\mathcal{L}_{\text{TR}}^t$, we ran gridsearch for each over possible values $[0.0, 0.01, 0.05, 0.08, 0.1, 0.15, 0.2, 0.4]$. We report the best settings across datasets in Table 9:

| | S-CIFAR-10 | S-CIFAR-100 | S-Tiny-ImageNet | P-MNIST | R-MNIST |
|---|---|---|---|---|---|
| $\alpha$ | 0.05 | 0.05 | 0.01 | 0.1 | 0.1 |
| $\beta$ | 0.01 | 0.01 | 0.01 | 0.05 | 0.05 |
| $\gamma$ | 0.2 | 0.08 | 0.05 | 0.1 | 0.1 |
| $\delta$ | 0.1 | 0.1 | 0.1 | 0.15 | 0.15 |

Table 9: Hyperparameters for loss contributions that were tuned on validation sets for each dataset.

# E    On the number of Monte Carlo samples

Figure 6 shows the effect of the number of Monte Carlo (MC) samples for global $N$ and task-specific $M$ latent variables on accuracy during inference. In particular, we observe two favorable spots in terms of accuracy, one centered around $(M = 1, N = 50)$ and the other around $(M = 10, N = 20)$. It is worth noting that the total number of inference time MC samples grow quadratically with the number of tasks $t$, *i.e.,* $\mathcal{O}(NMt)$, and that a higher number of samples leads to a larger computational overhead. For instance, based on Eq. (9) in the main paper, the inference on the 10-*th* task of S-CIFAR-100, *i.e.,* $t = 10$, given the two favorable spots amounts to selecting the set of task-specific module predictions with the least uncertainty from a total of (a) $1 \times 50 \times 10 = 500$ predictions using $(M = 1, N = 50)$, and (b) $10 \times 20 \times 10 = 2000$ predictions using $(M = 10, N = 20)$. We, therefore, opt for the more efficient setting of $(M = 1, N = 50)$ throughout our experiments in the paper.

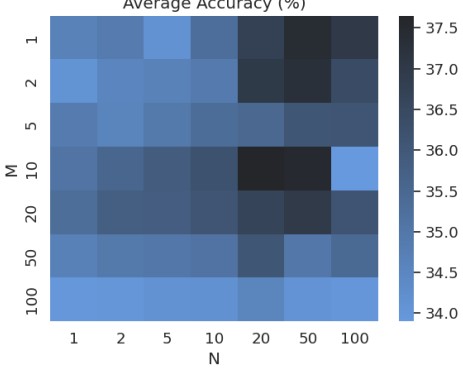

Figure 6: The effect of the number of Monte Carlo (MC) samples of global (N) and task-specific (M) latent variables on S-CIFAR-100 ($\mathcal{M}_{size} = 500$) accuracy.

| Method | S-CIFAR-10 Class-IL | | P-MNIST Domain-IL | | R-MNIST Domain-IL | |
|---|---|---|---|---|---|---|
| oEWC | -91.64 | | -36.69 | | -24.59 | |
| SI | -95.78 | | -27.91 | | -22.91 | |
| LwF | -96.69 | | - | | - | |
| $\mathcal{M}_{\text{size}}$ | 200 | 500 | 200 | 500 | 200 | 500 |
| ER | -61.24 | -45.35 | -22.54 | -14.90 | -8.24 | -7.52 |
| GEM | -82.61 | -74.31 | -29.38 | -18.76 | -11.51 | -7.19 |
| A-GEM | -95.73 | -94.01 | -31.69 | -28.53 | -19.32 | -19.36 |
| iCaRL | -28.72 | -25.71 | - | - | - | - |
| FDR | -86.40 | -85.62 | -20.62 | -12.80 | -13.31 | -6.70 |
| GSS | -75.25 | -62.88 | -47.85 | -23.68 | -20.19 | -17.45 |
| HAL | -69.11 | -62.21 | -15.24 | -11.58 | -11.71 | -6.78 |
| DER | -40.76 | -26.74 | -13.79 | -8.04 | -5.99 | -3.41 |
| ANP | -62.80 | -49.18 | -28.79 | -16.44 | -12.08 | -10.63 |
| ST-NPCL | -46.91 | -32.50 | -17.03 | -12.40 | -7.9 | -8.11 |
| **NPCL (ours)** | **-39.11** | **-27.62** | **-12.81** | **-8.60** | **-5.70** | **-4.10** |

Table 10: Backward transfer scores for the experiments in Table 1. Best results are in red. Second best results are in blue. All runs of the ANP, the ST-NPCL and the NPCL in the CL settings rely on experience replay (ER).

## F    Results: Backward Transfer

Table 10 reports the backward transfer for the accuracy scores mentioned in table 1. We further compute the backward transfer based on uncertainty scores to study the effect of forgetting on uncertainty. Fig. 7 shows the correlation between backward transfer of accuracy and uncertainty for the domain-IL datasets P-MNIST and R-MNIST.

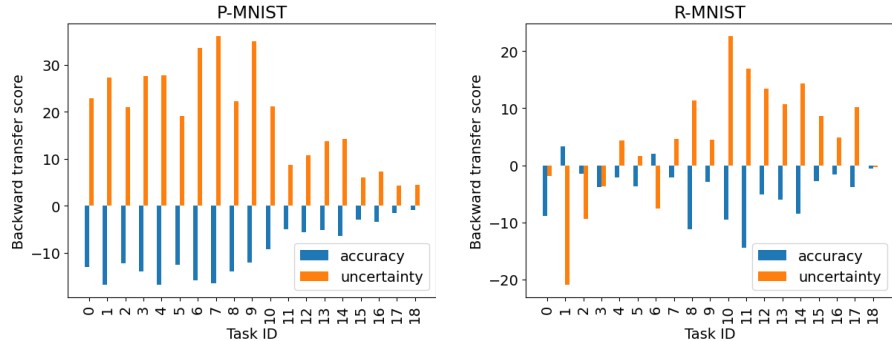

Figure 7: **Backward transfer scores of tasks based on accuracy and uncertainty on domain-IL datasets with** $|\mathcal{M}| = 500$: a higher negative backward transfer on accuracy correlates with a higher positive backward transfer on uncertainty and vice-versa. For better visibility, the uncertainty-based backward transfer scores have been scaled by a factor of 100.

## G    Ablations

### G.1    On the effect of regularization on the learned distributions

We record the per epoch L1-norms of global and task-specific means and variances on the last incremental task (task 4) of S-CIFAR-10. As shown in Fig. 8(a) and Fig. 8(b), regularizing the global distribution (GR) alleviates forgetting by limiting the learning of the global and the current task's (task 4) means and variances. This is evident through larger L1-norm of means and smaller L1-norm of variances when GR = 0, *i.e.,* +TR setting. On the other hand, excluding all the objectives, *i.e.,* the Baseline NPCL as well as excluding TR from the learning objectives, *i.e.,* +GR setting lead to

relatively unstable evolution of the past task means and variances, hence characterizing an increased forgetting. Including both GR and TR in the objective, *i.e.,* the NPCL helps find a balance between preserving the global and the past-task distributions while facilitating the learning of the current task distribution.

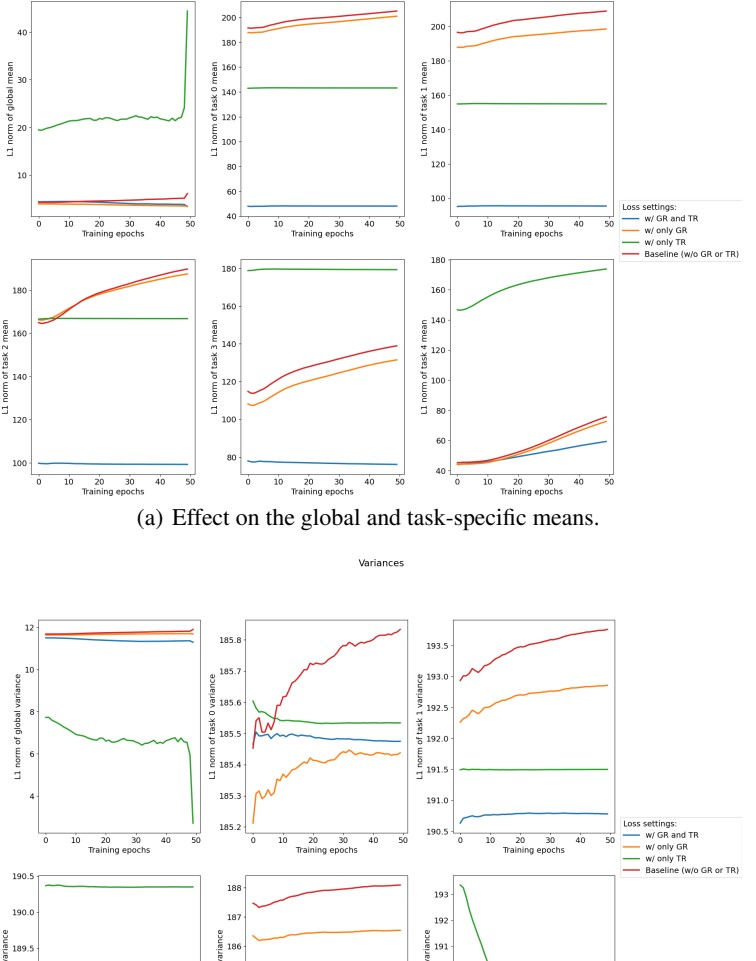

(a) Effect on the global and task-specific means.

(b) Effect on the global and task-specific variances.

Figure 8: Effect of the proposed global (GR) and task-specific (TR) regularizations on the learning of global and task-specific means and variances during the last incremental training task (task 4) of S-CIFAR-10. The NPCL uses both GR and TR while the baseline NPCL uses neither of them.

## G.2 How does forgetting effect uncertainty?

Fig. 9 ablates the average accuracies and uncertainties of each task head predictions over the test set of each task at the end of incremental training on S-CIFAR-100. Similar to S-CIFAR-10 (Fig. 4), we observe that the accuracy of predictions made by true task heads are higher than the rest. For predictive uncertainties, the trend is the opposite. Also, more recently trained tasks show lesser forgetting both in terms of accuracy (higher values) and uncertainty (lower values). This generalizes

our conclusion on S-CIFAR-10 regarding the outreach of forgetting in CL going beyond accuracy and to other aspects of learning such as the model's predictive confidence.

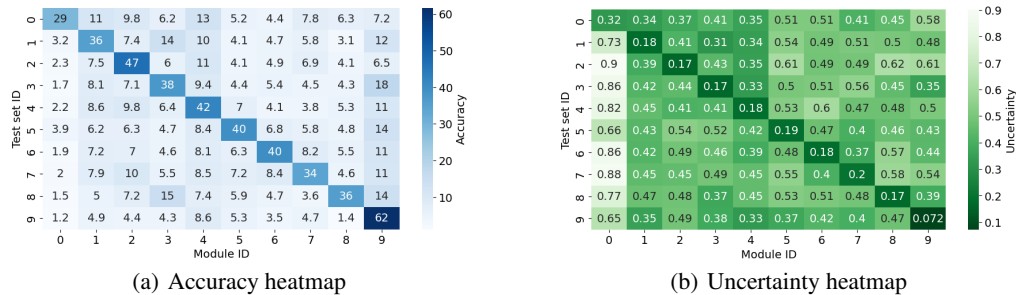

(a) Accuracy heatmap  (b) Uncertainty heatmap

Figure 9: Heatmaps depicting the average accuracy and uncertainty of individual task test sets per task head on S-CIFAR-100 with $|\mathcal{M}| = 500$ over an individual run.

### G.3 On the storage gain of the NPCL over DER

Table 11 compares the total episodic memory sizes of the NPCL (ours) and DER [4]. We report storage sizes as the dimension of a single 1-d vector constructed by flattening all the vectors that need to be stored by each method in the episodic memory. Namely, the NPCL stores $2 * t + 2$ vectors of fixed dimension $\mathbb{R}^{|o|}$ where $t$ is the total number of tasks in a dataset and $o$ is the output size of the mean and variance heads. On the other hand, DER stores $|\mathcal{M}|$ number of logits of dimension $\mathbb{R}^{|N_C|}$ where $N_C$ denotes the total number of classes in a CL dataset. As a result, the NPCL has significant storage gains on settings with either large number of classes or a larger memory size. It is worth noting that both the NPCL and DER rely on storing original input images and therefore, our comparison does not take the inputs into account.

| Method | S-CIFAR-10 | | S-CIFAR-100 | | S-Tiny-ImageNet | | P-MNIST | | R-MNIST | |
|---|---|---|---|---|---|---|---|---|---|---|
| $\mathcal{M}_{\text{size}}$ | 200 | 500 | 500 | 2000 | 200 | 500 | 200 | 500 | 200 | 500 |
| DER [4] | 2000 | 5000 | 50000 | 200000 | 40000 | 100000 | 2000 | 5000 | 2000 | 5000 |
| NPCL (ours) | 3272 | 3572 | 6132 | 7632 | 5832 | 6132 | 1544 | 1844 | 1544 | 1844 |
| Storage gain (%) | -63.6 | **28.56** | **87.746** | **96.184** | **85.42** | **93.868** | **22.8** | **63.12** | **22.8** | **63.12** |

Table 11: Storage size comparison of the NPCL with DER across different experimental settings of Table 1. Gains are marked in **bold**.

### G.4 On the importance of correct context

To study the significance of correct task-specific context $\mathcal{C}^t$ for the NPCL, we design a simple experiment. While training on an incremental task $t > 0$ onwards, after having derived the global latent samples $\{z^G\}_{i=1}^N$, we tinker with the flow of the task-specific context points $\mathcal{C}^t$ to the different task-specific encoders of the NPCL. In specific, instead of directing $\mathcal{C}^t$ to the $t$-th encoder, we *misdirect* it to the task encoder $j \ni j \neq t$ where, $j$ is chosen at random from the pool of all seen task ids. Note that the presence of such randomly allocated context points during training implies that we now have a noisy task-specific prior $q_\phi(z^t|z^G, \mathcal{C}^j)$ to match in the ELBO, *i.e.,* a corrupted second term on the right-hand side of eq. (5). We keep all our other training settings (including the loss coefficient values) unchanged.

Table G.4 compares the performance of the NPCL with the noisy task-specific priors on the two different memory sizes of S-CIFAR-10. While the performance gain of the NPCL over its noisy prior counterpart remains significant, we observe that in comparison with the ST-NPCL (which lacks hierarchy), the presence of noisy priors degrade the performance of the NPCL (which *has* hierarchy) further as the replay memory size increases from 200 to 500. This is because, with a larger memory size, more context points from past tasks are diverted to the random task components during training. This leads to a noisier task-specific prior matching. Such noisy priors further lead to higher

fluctuations in the accuracy, as marked by the larger standard deviations in their accuracy over the ST-NPCL and the NPCL. This **validates** the fact that the conditioning on the correct task-specific context remains crucial to the performance of the NPCL.

| Method | Average accuracy over 10 runs (%) | |
| --- | --- | --- |
| | S-CIFAR-10 ($\mathcal{M}_{\text{size}} = 200$) | S-CIFAR-10 ($\mathcal{M}_{\text{size}} = 500$) |
| ST-NPCL (w/ only per-task latent) | $54.6 \pm 2.14$ | $65.22 \pm 1.89$ |
| NPCL w/ noisy task-specific priors | $59.41 \pm 3.0$ | $65.1 \pm 2.31$ |
| NPCL | $63.78 \pm 1.7$ | $71.34 \pm 1.48$ |

Table 12: **The importance of correct context**: Comparison of S-CIFAR-10 accuracy (over 10 runs) of Single Task NPCL (ST-NPCL), the NPCL with noisy/corrupted task-specific priors and the standard the NPCL. The presence of noisy priors degrades the performance of the NPCL up to the extent of falling behind the ST-NPCL (without a hierarchical structure) on the setting with larger memory size.

## G.5    On out-of-the box novel data identification

Our novel data identification experiments use the S-CIFAR-10 and S-CIFAR-100 datasets interchangeably as $\mathcal{D}_{\text{ID}}$ and $\mathcal{D}_{\text{OOD}}$ given the high degree of similarity between a number of their classes [21].[2] Namely, while evaluating the NPCL trained on S-CIFAR-100, we consider the entire CIFAR-10 test set as $\mathcal{D}_{\text{OOD}}$ whereas the evaluation of the S-CIFAR-10 model treats the test set of first 10 class labels of CIFAR100 to be $\mathcal{D}_{\text{OOD}}$. Further, for an incremental task $t$, the test sets for $[0, t]$ tasks make up for the ID data $\mathcal{D}_{\text{ID}}$.

As shown in Table 13, the variances computed using either of our proposed metrics on $\mathcal{D}_{\text{ID}}$ are up to a magnitude lower than those on $\mathcal{D}_{\text{OOD}}$. This trend is evident across the incremental evaluation steps even if the differences in the variances between $\mathcal{D}_{\text{ID}}$ and $\mathcal{D}_{\text{OOD}}$ slump with the further arriving tasks. Moreover, for the model trained on the more challenging S-CIFAR-100 setting, we observe that the differences between the $\mathcal{D}_{\text{ID}}$ and $\mathcal{D}_{\text{OOD}}$ variances even grow during the course of incremental training. This implies the potential perks of enabling the inter-task knowledge sharing among the NPCL parameters in a CL setup.

| Incremental step | CIFAR-100 on S-CIFAR-10 model | | | | CIFAR-10 on S-CIFAR-100 model | | | |
| --- | --- | --- | --- | --- | --- | --- | --- | --- |
| | $\mathcal{D}_{\text{ID}}$ ($\delta$) | $\mathcal{D}_{\text{OOD}}$ ($\delta$) | $\mathcal{D}_{\text{ID}}$ (H) | $\mathcal{D}_{\text{OOD}}$ (H) | $\mathcal{D}_{\text{ID}}$ ($\delta$) | $\mathcal{D}_{\text{OOD}}$ ($\delta$) | $\mathcal{D}_{\text{ID}}$ (H) | $\mathcal{D}_{\text{OOD}}$ (H) |
| 1 | $1e^{-6}$ | $1e^{-5}$ | $9.3e^{-6}$ | $8.4e^{-5}$ | $1.5e^{-6}$ | $8.9e^{-6}$ | $1.5e^{-4}$ | $1e^{-3}$ |
| 2 | $2.6e^{-6}$ | $1.4e^{-5}$ | $6.3e^{-5}$ | $2.2e^{-4}$ | $1.9e^{-6}$ | $5.8e^{-6}$ | $5.3e^{-4}$ | $1.7e^{-3}$ |
| 3 | $2.3e^{-6}$ | $6.2e^{-6}$ | $6.7e^{-5}$ | $2.1e^{-4}$ | $1.2e^{-6}$ | $3.6e^{-6}$ | $4.4e^{-4}$ | $1.5e^{-3}$ |
| 4 | $8.1e^{-7}$ | $4.8e^{-6}$ | $4.6e^{-5}$ | $2.2e^{-4}$ | $1.1e^{-6}$ | $2.5e^{-6}$ | $3.5e^{-4}$ | $1.2e^{-3}$ |
| 5 | $7.1e^{-7}$ | $1.7e^{-6}$ | $4.6e^{-5}$ | $1.1e^{-4}$ | $8e^{-7}$ | $2e^{-6}$ | $4.4e^{-4}$ | $1.2e^{-3}$ |
| 6 | - | - | - | - | $6.8e^{-7}$ | $1.3e^{-6}$ | $4.1e^{-4}$ | $8.5e^{-4}$ |
| 7 | - | - | - | - | $4.e^{-7}$ | $1.3e^{-6}$ | $3.2e^{-4}$ | $8.3e^{-4}$ |
| 8 | - | - | - | - | $4.9e^{-7}$ | $1e^{-6}$ | $3.2e^{-4}$ | $6.7e^{-4}$ |
| 9 | - | - | - | - | $3e^{-7}$ | $6.9e^{-7}$ | $2.5e^{-4}$ | $4.7e^{-4}$ |
| 10 | - | - | - | - | $3.3e^{-7}$ | $5.1e^{-7}$ | $2.5e^{-4}$ | $3.5e^{-4}$ |

Table 13: Average variances over softmax ($\delta$) and entropy ($H$) scores of incremental models on in-distribution (ID) and out-of-distribution (OOD) test sets using $N = 50$ samples

## G.6    On instance-level model confidence evaluation

For each target instance $x_*$, the instance-level model confidence evaluation framework [17] uses the $N$ predictions obtained from stochastic sampling to compute: (a) the prediction interval width (PIW) between the $[2.5, 97.5]$ percentile range of the $N$ predicted classes, (b) the paired two-sample $t$-test [10] to evaluate the significance of difference between the mean predicted probabilities for the top-2

---

[2]The labels for first ten CIFAR-100 classes are the same as `https://huggingface.co/datasets/cifar100` and that for CIFAR-10 classes are the same as `https://huggingface.co/datasets/cifar10`.

most predicted classes. As a prerequisite to the latter test, we first verify the normality assumption of the probability differences for the NPCL (Fig. 10).

Similar to Fan et al. [10], after computing the PIW per test instance, we split the instances into two groups by the correctness of the majority-vote predictions, obtain the PIW of the true class per instance, and compute the mean PIW of the true class within each group. For t-test evaluation, we compute the mean accuracy per group of the test instances split by their $t$-test rejection status.

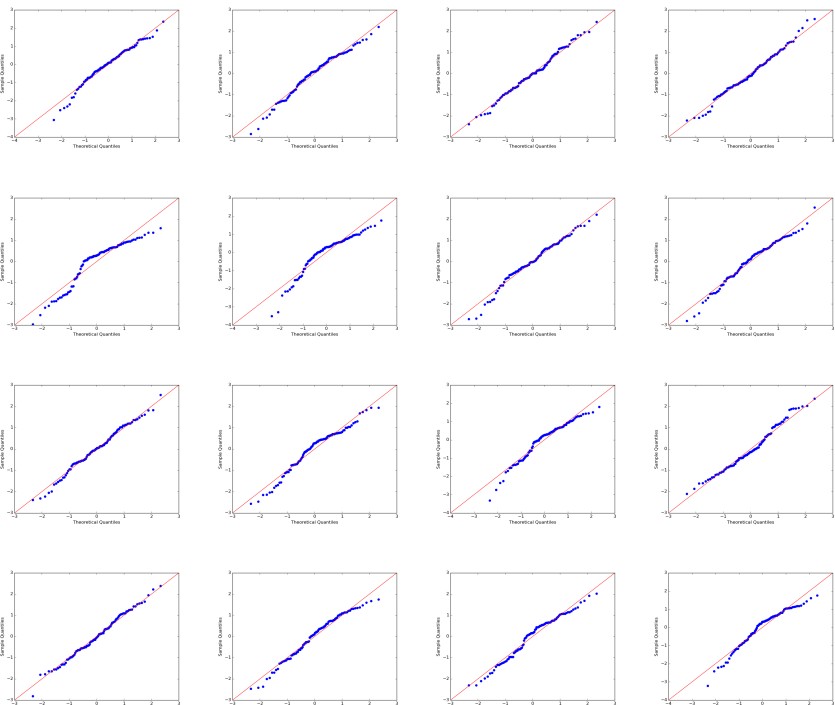

Figure 10: Q-Q plots for the differences in probability between the most and the second most predicted class.

| Class | Accuracy | PIW | | Accuracy by $t$-test status | |
| --- | --- | --- | --- | --- | --- |
| | | Correct | Incorrect | Rejected | Not Rejected |
| 1 | 82.30 | 74.17 | 102.21 | 83.37 | 50.00 |
| 2 | 94.00 | 62.90 | 79.86 | 94.07 | 80.00 |
| 3 | 74.00 | 54.92 | 68.48 | 74.14 | 64.29 |
| 4 | 71.50 | 65.42 | 74.32 | 72.06 | 25.00 |
| 5 | 84.80 | 92.93 | 106.90 | 85.37 | 22.22 |
| 6 | 76.50 | 75.22 | 103.58 | 76.58 | 60.00 |
| 7 | 94.20 | 104.9 | 129.56 | 94.39 | 3.00 |
| 8 | 90.50 | 81.10 | 127.06 | 91.12 | 22.22 |
| 9 | 96.90 | 72.81 | 110.86 | 97.00 | 66.67 |
| 10 | 96.30 | 80.60 | 109.56 | 96.48 | 60.00 |

Table 14: PIW (multiplied by 100) and $t-$test results for classes inferred from their respective task heads after S-CIFAR-10 training.

# H    Incompetence of Dot-product attention

Top-15 context points attended per query

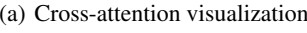

(a) Cross-attention visualization

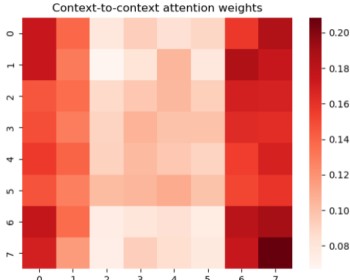

(b) Self-attention visualization

Figure 11: **Scaled dot-product attention visualization**: (a) top-15 context (buffer) points attended for 4 randomly chosen queries (test set samples). The queries are made after training on S-CIFAR-10. The sizes of the points correspond to the attention values while the colors denote the tasks they belong to. (b) self-attention weights of context points when all feature values are arranged in ascending order (along x and y-axis) shows that the ANP [27] mostly attends to the lowest or the maximum values in the context dataset.

