# NPCL: Neural Processes for Uncertainty-Aware Continual Learning

## Abstract

Continual learning (CL) aims to train deep neural networks (DNNs) efficiently on streaming data while limiting the forgetting caused by new tasks. However, learning transferable knowledge with less interference between tasks is difficult, and real-world deployment of CL models is limited by their inability to measure predictive uncertainties. To address these issues, we propose handling CL tasks with neural processes (NPs), a class of meta-learners that encode different tasks into probabilistic distributions over functions all while providing reliable uncertainty estimates. Specifically, we propose an NP-based CL approach (NPCL) with task-specific modules arranged in a hierarchical latent variable model. We tailor regularizers on the learned latent distributions to alleviate forgetting. We then use uncertainty estimation capabilities of NPCL to handle the fundamental CL challenge of task head inference. Our experiments show that NPCL outperforms previous CL approaches. We validate the effectiveness of uncertainty estimation in NPCL for identifying novel data and evaluating instance-level model confidence.

## 1 Introduction

Continual learning (CL) aims to help deep neural networks (DNNs) learn from a stream of non-stationary tasks by retaining the previously acquired knowledge [48, 32]. To achieve this, CL agents target alleviating the *catastrophic forgetting* issue with restricted computational and memory costs [37]. This requires balancing the plasticity for new knowledge with the stability for old [33].

To avoid forgetting in CL, experience replay (ER) methods [27, 6] are one effective way to train DNNs on a memory buffer with a subset of the past tasks' experiences. Other than the ER methods, many regularization-based approaches have also been proposed to penalize the forgetting on the DNNs' parametric [27] or representation spaces [5, 4]. However, these may still suffer from interference due to the regularization on the entire parameter space [7]. To address this, parameter isolation methods [47, 30] define task-specific training components but are usually confined to task incremental CL setups [42]. It is thus challenging for CL agents to maintain transferable and shareable knowledge. Furthermore, a hurdle to the real-world deployment of CL agents is their inability to measure predictive uncertainties. This, like other autonomous learning agents, keeps them from safety critical applications [28].

To tackle the above issues, we propose to explore CL models using neural processes (NPs) [12, 13], a class of meta-learners that model tasks as data generating functions from a stochastic process. NPs learn a prior over functions by marginalizing over a set of data points, or *context*, thus enabling rapid adaptation to new observations through inference on functions. Additionally, their probabilistic nature endows them with reliable uncertainty quantification capabilities [13, 24]. Our motivations to explore NPs for CL are thus two-fold. First, NPs exploit Bayes' theorem which naturally enables CL through sequential posterior construction. Namely, NPs perform inference over the function

space by learning context-based priors which are updated to posteriors upon observing (additional) *targets*. Second, NPs meta-learn input correlations through a latent variable. For CL, this could be the key to meta-learn knowledge transfer across correlated tasks. However, NPs cannot directly handle CL tasks given: (a) the reliance on a single global latent leads to suboptimal modeling of complex CL signals where multiple correlated tasks could occur simultaneously, (b) NPs still need to handle the forgetting of past task correlations arising from the non-static data stream.

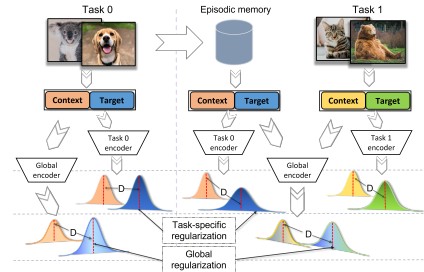

To address the above desiderata, we propose Neural Processes for Continual Learning (NPCL), a hierarchical latent variable model with a global latent to capture inter-task correlation and task-specific latents for finer knowledge. Fig. 1 shows NPCL exploiting functional correlation among current and past task training samples of ER. The drift of global and past task-specific distributions away from their original forms serve as the major aspect of forgetting in NPCL. We thus propose to regularize these towards their old forms and show the merits of it over typical parameter-based regularization. We then leverage the uncertainty encoded by NPCL for the aforesaid CL challenge of task head inference. To this end, we propose using entropy as an uncertainty quantification metric (UQM). NPCL outperforms previous probabilistic CL models and delivers better or comparable results than state-of-the-art deterministic CL methods, which usually have an edge over their probabilistic counterparts in terms of accuracy. To study the further usages of

Figure 1: **Neural Processes for Continual Learning**: each training step involves minimizing the distance $D$ between the context-based prior and the target-based posterior, alongside regularizing the task-specific and global distributions towards their old forms.

NPCL's uncertainty estimation, we show its out-of-the-box readiness for novel data detection and instance-level confidence evaluation [16]. Lastly, we list the key limitations of NPCL as an attempt to lay further solid directions for uncertainty-aware CL.

## 2 Related Work

**Continual Learning (CL).** CL methods address catastrophic forgetting through three major approaches: (a) *Regularization-based* methods penalize changes in a model's important weights for previous tasks; e.g., Elastic Weight Consolidation (EWC) [27], Synaptic Intelligence (SI) [48], etc. (b) *Parameter Isolation-based* methods partition the network's parameters to specialize on individual tasks; e.g., Yan et al. [46] using variational Bayesian sparsity priors to reserve model capacity for future tasks, Douillard et al. [8] learning task-specific tokens for Transformers, etc. (c) *Replay-based* methods use an episodic memory to preserve a fraction of the past tasks' experience, and use these to prevent forgetting while learning on new tasks; e.g., experience replay (ER) [6] storing past inputs, and dark experience replay (DER) [4] storing past logits. Our method uses (a) via regularization of distributions, (b) via task-specific latent heads, and (c) via replay of past task inputs and distributions.

**Neural Processes (NPs).** NPs were introduced to meta-learn a family of data-generating functions through their deterministic [12] and / or latent summaries [13]. Attentive NPs (ANPs) [24] replaced the averaging operation in NPs with a dot-product attention [43] to enhance their expressivity. (A)NPs rely on a global latent that limits their ability to model observations from multiple functions. Recent works address this through local latents that model fine-grained correlation among a subset of the observations [45]. In particular, our work is inspired by multi-task processes (MTPs) [23] that model multiple tasks owing to the hierarchy of task-specific latents conditioned on a global latent. However, MTPs have more relaxed constraints than CL because: (a) MTPs are not trained sequentially on tasks and are thus free of forgetting; (b) at test time, MTPs assume each input to be mapped to all tasks and thus bypass the issue of task head inference pervading class-incremental learning.

Besides, the added complexity of variational inference has limited NP applications to mostly toy regression tasks [22]. The potential of NPs for large-scale classification tasks thus remains largely unexplored, if not untouched. Wang et al. [44], for instance, leverage the predictive uncertainties of NPs to decide on pseudolabels for unlabeled data in semi-supervised classification. We, therefore, use NPs for CL because of a number of their intrinsic properties including principled Bayesian learning, uncertainty estimation, and easy integration with preexisting CL methods like ER.

## 3 Preliminaries: Neural Processes

NPs [12, 13] meta-learn a task $t$ as the mapping $F_*^t : X^t \to Y^t$. $F_*^t$ generates data $\{(x^t, y^t)\} = (X^t, Y^t) \sim \mathcal{D}^t$ constituting of: (a) a context set $|\mathcal{C}^t| = m$ that offers a prior, and (b) a target set $|\mathcal{T}^t| = m + n$ that contains additional samples to compute the posterior over the full observations. NPs learn the Gaussian priors and posteriors using a neural network $F_{[\phi;\theta]}^t \approx F_*^t$, where $\phi$ and $\theta$ parameterize an encoder $q$ and a decoder $p$, respectively. This involves deriving a global variable $z^G$ to estimate the prior $p(z^G|\mathcal{C}^t; \phi)$, and then maximizing the marginal likelihood $p(Y_{\mathcal{T}}^t|\mathcal{C}^t, X_{\mathcal{T}}^t; \theta)$:

$$p(Y_{\mathcal{T}}^t|X_{\mathcal{T}}^t, \mathcal{C}^t) = \int p(Y_{\mathcal{T}}^t|X_{\mathcal{T}}^t, z^G)p(z^G|\mathcal{C}^t)dz^G, \tag{1}$$

where $p(Y_{\mathcal{T}}^t|X_{\mathcal{T}}^t, z^G) = \prod_{i=1}^{m+n} p(y_i^t|x_i^t, z^G)$ is the generative likelihood. In a CL setup, memorizing the task prior $p(z^G|\mathcal{C}^t; \phi)$ can help NPs avoid forgetting the $t$-*th* task. Our aim behind enabling NP for CL is to seek a trade-off to preserve such task priors while sharing the parameters among tasks.

## 4 Continual Learning with Neural Processes

A CL setup considers $\mathcal{D}^t$ from $0 \le t \le T - 1$ sequentially arriving tasks. Using cross-entropy (CE) as the classification loss $l$, the CL objective for the task $t$ involves minimizing:

$$\mathcal{L}_{CE}^t = \mathbb{E}_{(x,y) \sim \mathcal{D}^t} \, l(F_{[\phi;\theta]}(x), y) \tag{2}$$

on all $[0, t]$ seen tasks. Achieving Eq. (2) is challenging in real-world scenarios where the previous datasets can be unavailable due to constraints on privacy, storage, etc. To bypass this, several CL approaches employ *experience replay* (ER) where a small episodic memory $\mathcal{M}$ is updated periodically to store and revisit minibatches of past experiences $(x^t, y^t) \sim \mathcal{M}$ for a task $t$ [6, 32]. In this work, we rely on reservoir sampling [6] for a task boundary-agnostic updating of $\mathcal{M}$ .

Jointly optimizing parameters on $\mathcal{D}^t$ and $\mathcal{M}$ has several drawbacks [32, 4]. From a network capacity view, exhausting the parameter space early makes interference from the latter tasks more likely. On a generative stand, the deterministic mapping $F^t$ limits capturing the randomness behind the real-world data. Owing to these, we next propose extending Eq. (2) with Eq. (1) to arrive at a CL model that: (a) allocates minimal parameters to learn robust per-task and global priors, and (b) uses generative factors to meet data-driven challenges such as deducing the right parameters for inference.

### 4.1 Neural Processes for Continual Learning

We begin with a direct extension of the NP formulation to CL. In an ER setup, where the context and target could be from $t$ tasks, Eq. (1) can be extended to derive the joint posterior for NPs [13] as:

$$p(Y_{\mathcal{T}}^{0:t}|\mathcal{C}^{0:t}, X_{\mathcal{T}}^{0:t}) = \int p(Y_{\mathcal{T}}^{0:t}|X_{\mathcal{T}}^{0:t}, z^G)p(z^G|\mathcal{C}^{0:t})dz^G, \tag{3}$$

where $z^G$ models the joint distribution $F_*^{0:t}$ of CL tasks and is an enabler of the knowledge transfer [32] between these (see App. A.3 for ELBO). Eq. (3) still poses two challenges. First, it needs the labeled context $\mathcal{C}$ for inferring predictions, which is impossible in the CL setups where test data are assumed to be unlabeled. To overcome this, we turn to using the memory $\mathcal{M}$ offered by the ER-based setups as *context* during inference. Second, jointly modeling $F_*^{0:t}$ ignores the dynamics of per-task stochasticities, and is still prone to the bottlenecks of Eq. (2). Addressing the latter, we next consider *redefining* Eq. (3).

### 4.2 NPs with Hierarchical Task-specific Priors for CL

To learn informative task priors with knowledge transfer, we presume two solid directions. First, inducing the knowledge transfer implies that we preserve the global latent $z^G$. Second, the task priors could be captured better if modeled explicitly. We thus extend Eq. (3) with task-specific latents $z^t = (z^0, .., z^t)$. As a result, our posterior is a two-step hierarchical latent variable model (Fig. 2) where the global and the per-task latents model the inter and intra-task correlations, respectively:

$$p(Y_{\mathcal{T}}^{0:t}|X_{\mathcal{T}}^{0:t}, \mathcal{C}^{0:t}) = \int \int \Big[ \prod_{t=0}^{T-1} p(Y_{\mathcal{T}}^t|X_{\mathcal{T}}^t, z^t)p(z^t|z^G, \mathcal{C}^t) \Big] p(z^G|\mathcal{C}^{0:t})dz^{0:t}dz^G, \tag{4}$$

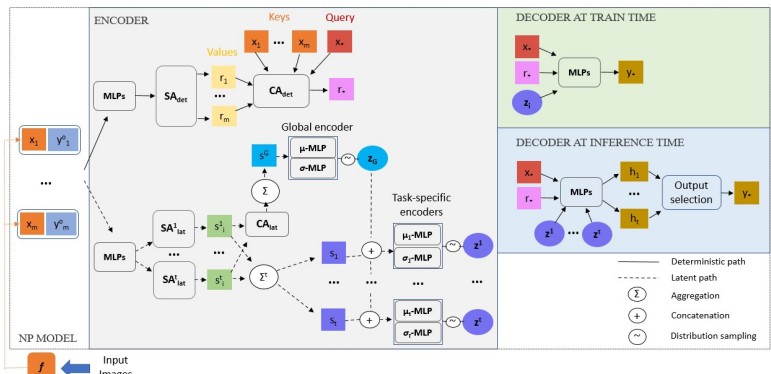

Figure 2: **NPCL architecture:** the decoding mechanism differs during training and inference.

where the entire context $\mathcal{C}^{0:t}$ is first encoded into $z^G$ and then conditioned on $z^G$, the task-specific context $C^t := (C^0, .., C^t)$ are encoded into their respective latents. We refer to Eq. (4) as NP for CL (NPCL). NPCL generalizes to the MTP [23] when the input labels span the entire output spaces (see section 2). Next, we detail the neural network architecture for NPCL.

### 4.3  NPCL Architecture

Given the input images $x_i \in \{\mathcal{C}, \mathcal{T}\}$, we first pass these to a feature extractor $f$. With a slight abuse of notation, we denote the features as $x_i : x_i \in \mathbb{R}^{|f|}$ from here onward. $x_i$ concatenated with the one-hot encoded labels $[x_i; y_i]$ is fed to the NPCL encoder with a deterministic and a latent path, and then to the decoder. All NPCL layers use multi-layer perceptrons (MLPs) projections, *i.e.,* $\mathrm{MLP}(x) : \mathbb{R}^{|f|} \rightarrow \mathbb{R}^{|o|}$ where, $o$ is a hyperparameter. We denote a normal distribution with a mean $\mu$ and a variance $\sigma^2$ by $\mathcal{N}(\mu, \sigma^2)$; the global and the task-specific distributions are $\mathcal{N}(\mu_G, \sigma_G^2)$ and $\mathcal{N}(\mu_t, \sigma_t^2)$. Lastly, by attention, we refer to the multi-head dot-product operations [43].

**Latent Encoder.** The latent path comprises of the projection $\Phi_i^{\mathrm{lat}} = \mathrm{MLP}([x_i; y_i])$ followed by two attention operations. First, per-task projections form the keys, values and queries to taskwise self-attention layers $SA_{lat}^t$ that produce order-invariant encodings $s_i^t$ over the samples of task $t$. Second, all encodings $\{s_i^{0:t}\}_{i=1}^{n+m}$ serve as the keys, values and queries to cross-attention layers $CA_{\mathrm{lat}}^{0:t}$ that enrich their order-invariance from intra-task $s^t$ to inter-task $s^G$. $s^t$ and $s^G$ are used to derive the $N$ and $M$ Monte Carlo samples of the global $z^G$ and the task-specific latents $z^t$, respectively (see App. B for more details) using the reparameterization trick [26]. We set $M = 1$ to enhance the inter-task stochasticity in posterior. For each input, we thus get $N * (t + 1)$ latent outputs.

**Deterministic Encoder.** The deterministic path is similar to that of the ANP [24] and outputs an order-invariant representation $r_*$ for target $x_*$ (see App. B).

**Decoder.** Based on the task information, the decoder adopts separate mechanisms during training and inference. At train time, we use the available task labels to filter the $N$ true latents $\{z_i^t\}_{i=1}^N$, combine them with $r_*$ and $x_*$, and decode the logits $h_*$. We discuss the inference time decoding in sec. 4.5.

### 4.4  Learning Objectives for NPCL

The learning of NPCL is done by variational inference that maximizes the evidence lower bound with

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

 ResNet-based ER as the NPCL layers are liable to more forgetting. Including TR into our objectives leads to the singlemost gain over the baseline. We further study how these objectives guide the learning of the global and task-specific means and variances with training (see App. E.1). We observe that NPCL w/ TR leads to better

| Method | S-CIFAR-10 | S-Tiny-ImageNet |
|---|---|---|
| ER | 44.79 | 8.49 |
| Baseline (w/o GR or TR) | 32.24 | 7.15 |
| NPCL (w/ only GR) | 50.68 | 8.61 |
| NPCL (w/ only TR) | 57.28 | 11.36 |
| NPCL (w/ GR and TR) | **63.78** | **12.44** |

Table 2: Accuracy w/ learning objectives

learning of the current task as well as preserving the past task distributions but at the cost of drifting the global distribution. NPCL w/ GR restricts the global distribution drift but not for the per-task distributions. NPCL w/ GR and TR strikes a balance in between.

**On Uncertainty.** Fig. 4 ablates the average accuracies and uncertainties of each task head predictions over the test set of each task on S-CIFAR-10 (see App. E.2 for S-CIFAR-100). First, we observe that the accuracy of predictions made by true task heads are, in general, a magnitude higher than the rest. For uncertainty, this trend is reversed. This verifies our assumption that restricting latent heads to learn only their true label distribution makes them more confident in modeling the within-task samples. Second, for recently trained tasks, the uncertainty differences between the true task heads and the rest are greater than the earlier tasks. This suggests that the extent of forgetting goes beyond a model's accuracy and to other aspects of learning such as its confidence. To further verify this, we probe the BWT of uncertainty, and see a strong correlation with the BWT of accuracy (see Fig. 7).

**On Context Size.** We study the average accuracy (Fig. 5(a)) and uncertainty (Fig. 5(b)) after training on S-CIFAR-10 with $|\mathcal{M}| = 200$, and then varying the context sizes during inference. Similar to other NPs [44, 11], we find a positive correlation between context size and performance indicating that NPCL utilizes useful information from diverse context, thereby reducing its task inference ambiguity.

**On Storage Overhead.** For each task, NPCL stores two new vectors – task-specific mean and variance, and replaces the global mean and variance with the current global ones. The NPCL storage thus scales constantly in the size $|\mathcal{M}|$ of the memory. This offers a strong edge on storage efficiency when compared to the SOTA [4] scaling quadratically, i.e., $|\mathcal{M} * N_C|$ where $N_C$ is the total number of classes. For instance, on S-Tiny-ImageNet with $|\mathcal{M}| = 500, |N_C| = 200$, NPCL's cumulative storage amounts to a (flattened) vector of size 6132 (256*10*2 for 256-d means and variances of 10 tasks + 256*2 for 256-d global mean and variance + 500 for 1-d task labels) while that of DER

amounts to 100,000 (200*500 for logits of 500 memory samples), *i.e.,* **a 93.868% storage gain**. We report the storage gains of NPCL over DER across all settings in App. E.3.

## 5.4 Applications of Uncertainty Quantification

The probabilistic nature of NPCL offers it an edge at leveraging data-driven UQMs. To further study the usage its predictive uncertainties, we design two experiments that leverage pretrained NPCL.

**Novel Data Identification.** Novel data identification seeks to distinguish out-of-distribution data ($\mathcal{D}_{OOD}$) from in-domain data ($\mathcal{D}_{ID}$). Forgetting makes CL models struggle further on the task [18]. The probabilistic sampling in NPCL opens the door for leveraging its predictive variances – which are more reliable estimates of aleatoric uncertainty than pointwise predictions [21]. For the $N$ predicted logits, we thus compute the variances over their softmax scores, $\sigma^2(\delta(h_*))$, and their uncertainty scores, $\sigma^2(U(h_*))$. Table 3 evaluates these met-

| Incremental step | $\mathcal{D}_{ID}$ = CIFAR-10, $\mathcal{D}_{OOD}$ = CIFAR-100 | | | |
|---|---|---|---|---|
| | $\mathcal{D}_{ID}$ ($\delta$) | $\mathcal{D}_{OOD}$ ($\delta$) | $\mathcal{D}_{ID}$ (H) | $\mathcal{D}_{