# OpenReview forum: "NPCL: Neural Processes for Uncertainty-Aware Continual Learning"
_NeurIPS.cc/2023/Conference — NeurIPS 2023 poster_

### Official Review · Reviewer_1ufQ · 2023-06-30

**Soundness:** 2 fair
**Presentation:** 3 good
**Contribution:** 2 fair
**Rating:** 4
**Confidence:** 4

**Summary:**

This paper proposes a new method for Continual Learning (CL) through Neural Processes (NPs) framework. Especially, they are inspired by MTP [1] which utilizes the global latent and task-specific latent for inter-task and intra-task knowledge representation. In this paper, the authors additionally regularize on parameter updates of MTP not too much be updated, and task-wise inference through an uncertainty quantification metric. Through this, they can achieve comparable performance with state-of-the-art prior works while maintaining smaller memory.

[1] Buzzega, Pietro, et al. "Dark experience for general continual learning: a strong, simple baseline." Advances in neural information processing systems 33 (2020): 15920-15930.

**Strengths:**

- This paper proposes new regulations to utilize NPs for CL.
- The prior works are well summarized and they discussed their model adequately.
- They evaluated it on multiple benchmarks including class and domain incremental learning benchmarks with diverse prior works including NPs [1] and ANPs [2], and showed their model, the NPCL shows comparable performance to state-of-the-art methods such as DER [3].
- They studied diverse ablations such as their regularization losses, latent types (e.g., when only task-wise latent is learned), task-wise inference, and context sizes. In this study, they showed their task-wise inference is important for their model's performance and the uncertainty on the logits of the task heads is a good measurement for task label estimation. Additionally, they also compared their model with DER for the storage overhead. For this aspect, NPCL shows more efficient storage utilization than DER.
- They also showed the uncertainty on the task heads can be utilized to identify the novel data from out-of-distribution or not and uncertainty-based task-wise inference can show higher confidence for correct task labels.
- Lastly, they clearly discussed the limitations of their model such as more computational overhead and inference time complexity caused by multi-head attention.

[1] Garnelo, M., Schwarz, J., Rosenbaum, D., Viola, F., Rezende, D. J., Eslami, S. M., & Teh, Y. W. (2018). Neural processes. arXiv preprint arXiv:1807.01622.

[2] Kim, H., Mnih, A., Schwarz, J., Garnelo, M., Eslami, A., Rosenbaum, D., ... & Teh, Y. W. (2019). Attentive neural processes. arXiv preprint arXiv:1901.05761.

[3] Buzzega, P., Boschini, M., Porrello, A., Abati, D., & Calderara, S. (2020). Dark experience for general continual learning: a strong, simple baseline. Advances in neural information processing systems, 33, 15920-15930.

**Weaknesses:**

- The motivation is not strong enough. Their motivation to utilize NPs for CL is to measure predictive uncertainties. However, it is not clearly discussed why it is important for CL.
- Figures (Figures 1, 2, 4, and 6) are not recognizable. Especially their fonts are too small. For Figure 6, they missed labeling the axes.
- In lines 38-39, they mentioned one of their motivations is NPs can meta-learn input correlations across correlated tasks. However, in the limitation section, they analyzed it as the incompetence of dot-product attention (lines 305-306). It is the opposite analysis of their motivation.
- The equation (7) and relevant explanations are confusing. In the previous section for GR, task $t$ is dominated by the t-$th$ task samples, while it is referred to as $j$ in this section. As I understood, they tried to regularize the task $t$'s task latent on that at the step when the task $t$ arrived, but it is not easy to follow in their explanation.
- The equation (8) is hard to understand. In their explanation about the memory management of their model, the sample $x$ is not stored, but it is not in this equation (without $x$, $L_{CE}$ cannot be calculated).
- The comparison in Table 1 can cause confusion. At first glance, we can expect NPCL and DER to show comparable performance in the same memory usage, but their storage usages are quite different as shown in Appendix E.3. I can recommend showing as a plot in which the x-axis is real storage usage and the y-axis is accuracy.
- (minor) typos

    - In equation (8), the bracket for $L_{GR}$ and $L_{TR}$ is missed.
    - In table 2, baseline is not w/o GR *or* TR, but GR *and* TR.
    - In line 208, with. -> with


**Questions:**

- Can you additionally analyze why small $N$ is not working well with the additional experimental results for diverse $N$ values?
- In your design, the sum pooling is used to get the global and task-specific latent. Could you replace it with Multi-head Attention Pooling (MAP) [1] (e.g., using the learnable additional token for the latent) and test it for a few tasks?

[1] Lee, J., Lee, Y., Kim, J., Kosiorek, A., Choi, S., & Teh, Y. W. (2019, May). Set transformer: A framework for attention-based permutation-invariant neural networks. In International conference on machine learning (pp. 3744-3753). PMLR.


**Limitations:**

The authors adequately discussed the limitations of their work. Thanks for their detailed analysis of their limitations.

---

> ### Author Rebuttal · Authors · 2023-08-09
>
> We thank the reviewer for their careful reading and remarks on our work. In regard to the questions raised, we provide our explanations below:
>
> - **Motivation**: Our motivation to utilize NPs for CL is two-fold. First, based on lines 20-26, we aim to design parameter isolation methods for CL that can better: (a) exploit task-specific and inter-task correlations, and (b) infer which task-specific component to utilize for test samples in the lack of a given task id.  Second, we are interested in CL models that are reliable in their predictive confidence. This latter goal is particularly useful for the real-world deployment of models where changing data distributions can interfere with a model's interpretability of its predictions. We thus wish to know when to rely on a model’s predictions and when to call for human/expert interventions. We will elaborate further on our motivation in the next version of our paper.
>
> - **Lines 38-39**: Yes, it is true that NPs can meta-learn input correlations across correlated tasks. However, the degree of the learned correlations can vary with the type of inductive bias used by the model [1]. For instance, the attentive NPs using dot-product attention capture better correlations than the vanilla NPs. Nevertheless, dot-product attention, like other optimization algorithms, suffers from its own limitations [2]. Rather than an opposite analysis of our motivation, we thus suggest that lines 305-306 be read as an implication of the no-free-lunch theorem for supervised learning [3].
>
> - **On Eq. (7)**:  We apologize for the confusion and provide further explanation on the notation here. In line 165, the task $t$ indeed refers to the current training task whose samples dominate the training set (given that the replay memory $\mathcal{M}$ contains a much smaller number of samples per past task). On line 172, by task $t$, we actually refer to the past tasks for which we are regularizing the learned distribution (learned from $\mathcal{M}$) towards the distribution that we had learned when the past task was first seen, *i.e.,* the step $0 <= j < t$. To retrieve the latter past task distribution, we rely on the distribution memory $\mathcal{M}_\mathcal{N}$ (line 173).
>
> - **On Eq. (8)**: We believe there has been a misunderstanding regarding how the replay memory $\mathcal{M}$ works. $\mathcal{M}$ does  store the sample $x$. We state this very clearly in lines 107-109, and also in appendix E.3 (lines 624-625).  It is in addition to this memory $\mathcal{M}$ that NPCL stores the *separate* distribution memory (line 173). Given that $x$ is available from $\mathcal{M}$, $\mathcal{L}_{CE}$ can thus be calculated.
>
> - **Table 1**: Following the previous point, we reiterate the fact that both NPCL and DER store the original sample $x$ in the replay memory $\mathcal{M}$ (lines 107-109; Appendix E.3 lines 624-625). The difference in memory usage is that while DER relies on storing the logits for the individual samples $x$, NPCL instead stores the parameters of the global and task-specific distributions as well as the scalar task id of $x$. Given that both NPCL and DER store sample $x$, we do not take these into account for our comparison of storage gain (lines 624-625). We nevertheless agree with the idea of a plot depicting the real storage usage and plan to incorporate this.
>
> - **Analysis of $N$ values**: Please refer to the global response and the Fig. 3(a) in the attached PDF for the analysis. Intuitively, the hierarchy in NCPL implies that the $M$ Monte Carlo (MC) samples of different task-specific distributions are conditioned on the same set of $N$ MC samples of the global distribution (see Fig. 3(b) for a rough sketch of this).  With smaller $N$, we restrict the diversity of the samples $z^G$  of the global distribution that the task-specific encoders should look at. This is equivalent to limiting the support of the global distribution. We know that different regions of the global distribution could capture different CL tasks. Smaller $N$ values are thus likely to condition a task $t$'s encoder on a global sample $z^G$ that represents a region corresponding to another task that is less correlated to $t$. Similarly, with smaller $N$, we are also more likely to miss out on conditioning the $t-th$ encoder on samples from regions of the global distribution that capture tasks having a high correlation with $t$.
>
> - **Multi-head Attention Pooling (MAP)**: Per your advice, we experiment with using Multi-Head Attention Pooling (MAP) [4] as a possible alternative to the global average pooling. The results of NPCL with MAP on S-CIFAR-10 with a memory size of 200 are shown below. In particular, we observe that while MAP does lead to a slight improvement of accuracy on the first task, the additional number of learnable parameters inadvertently causes higher forgetting on subsequent incremental tasks. We thus leave the exploration of MAPs for NPs in CL as a potential direction for future work.
>
> | Method        | Task 1 | Task 2 | Task 3 | Task 4 | Task 5 |
> |---------------|--------|--------|--------|--------|--------|
> | NPCL          | 98.3   | 86.12  | 74.85  | 69.48  | 63.78  |
> | NPCL w/ MAP   | 98.46  | 80.43  | 68.2   | 59.74  | 51.77  |
>
> **References**:
>
> [1] Jha, S., Gong, D., Wang, X., Turner, R.E., & Yao, L. (2022). The Neural Process Family: Survey, Applications and Perspectives. *ArXiv, abs/2209.00517*.
>
> [2] Kim, M., Go, K., & Yun, S. (2022). Neural Processes with Stochastic Attention: Paying more attention to the context dataset. *ICLR*.
>
> [3] Goldblum, M., Finzi, M., Rowan, K., & Wilson, A.G. (2023). The No Free Lunch Theorem, Kolmogorov Complexity, and the Role of Inductive Biases in Machine Learning. *ArXiv, abs/2304.05366*.
>
> [4] Lee, J., Lee, Y., Kim, J., Kosiorek, A., Choi, S., & Teh, Y. W. (2019, May). Set transformer: A framework for attention-based permutation-invariant neural networks. In International conference on machine learning (pp. 3744-3753). PMLR.

---

> > ### Comment · Reviewer_1ufQ · 2023-08-14
> > **Reply to the response of the author**
> >
> > Thank you for your response to my concerns. I think they properly discussed them.
> >
> > +Can you update the figures? They are hard to understand due to very small fonts and missed x and y axis labels.

---

> > > ### Author Response · Authors · 2023-08-14
> > >
> > > Dear Reviewer 1ufQ,
> > >
> > > Thank you for taking the time to review our rebuttal. We have considered your feedback regarding the figures and plan to update our manuscript at the earliest allowed revision.
> > >
> > > Let us know if you have any other questions or concerns. Also, if you are satisfied with our answers, please consider revising your score.
> > >
> > > With best regards

---

> > > > ### Author Response · Authors · 2023-08-17
> > > >
> > > > Dear Reviewer 1ufQ,
> > > >
> > > > We have put our best efforts in attempting to address most of your concerns by providing more reasonable explanations and/or new experimental results.
> > > >
> > > >  If our response addresses your concerns and clarifies the value of our paper, we'd be truly grateful if you could reconsider your recommendation.
> > > >
> > > > Please be assured that we are open and eager to continue the discussion should there be any aspects of our paper that remain unclear to you. Thank you once more for taking the time to engage with our response!
> > > >
> > > > With best regards

---

### Official Review · Reviewer_v3Yx · 2023-07-05

**Soundness:** 2 fair
**Presentation:** 2 fair
**Contribution:** 3 good
**Rating:** 6
**Confidence:** 4

**Summary:**

This paper suggests tackling continual learning (CL) with a neural process model (NP).  The authors describe a hierarchical latent variable model, with one global latent, and t per-task latents.  As is common in NPs, the latent posteriors are a function of a context of datapoints with their corresponding labels, and then a decoder maps the latents to an output prediction.

During training time, the model is trained with access to task labels, and uses data from different tasks to train the corresponding task-specific latent posteriors. A replay buffer stores and reuses data from old tasks during training, and additional regularization terms are introduced, in order to minimize effects of drifting away from previous tasks ('forgetting').

At test time, there is no access to task labels, and therefore the model is run for all tasks, and the output with the least entropy is used.

The model is compared to various CL approaches and is shown to achieve and sometimes surpass the state of the art. The paper describes more ablations and analysis of the results.

**Strengths:**

The main strengths of this paper are:
1. An interesting model with non-trivial components put together:
    - the hierarchical structure of the latents
    - the difference treatment of training time and test time with respect to access to context latent
2. The result seems to surpass previous approaches
3. Extensive analysis of the results

**Weaknesses:**

The main weakness of this paper is that modeling continual learning as meta learning via neural processes is not exactly valid.
Specifically, NPs assume that the dataset contains many different functions or tasks, and one of the main challenges is that the task id is not accessible and has to be inferred. This is the reason a "context" of labeled points is used - to infer the task at hand.
In the setup described in the paper, it is assumed that the task id is accessible at training time, which allows designating different parts of the model to different tasks.
At test time, the model uses a context coming from the training data with little overlap to the task of the test data. This is a major difference from NPs.
I think that the presented model is interesting and can have a significant contribution to the community but that the good performance exhibited is not connected to NPs as stated, but rather merely a result of the higher capacity of hierarchical latent variable models.
Since the model is conditioned on training points during evaluation, perhaps a better way to present it is as a non-parametric method (e.g.[1]) rather than a neural process.

[1] Self-Attention Between Datapoints: Going Beyond Individual Input-Output Pairs in Deep Learning. Jannik Kossen, Neil Band, Clare Lyle, Aidan N. Gomez, Tom Rainforth, Yarin Gal

**Questions:**

1. The paragraph in line 110 is not clear. What does it mean to optimize parameters of $\mathcal{D}^t$?
2. In equation 3, does the context define the task (as modeled in standard NPs)? If so why are there superscripts of all the tasks 0:t?
3. In equation 4 why are $z^t$ conditioned on the contexts $\mathcal{C}^t$? If the task id is given then the context doesn’t have any information.
4. The number of samples described in line 151 is not clear. If every point is mapped to one task then only the corresponding latent needs to be sampled, making the number of samples M. Is this wrong?

**Limitations:**

I did not find any unaddressed limitations or potential negative impact in this work.

---

> ### Author Rebuttal · Authors · 2023-08-09
>
> We thank the reviewer for their careful reading and remarks on our work. In regard to the questions raised, we provide our explanations below:
>
> - **Validity of NPs for CL**: We believe that the remark against the validity of our setting stems from a misunderstanding between the standard meta-learning (ML) settings (that NPs were initially designed for) and the continual learning (CL) setting (ours). Namely, in ML testing, we are interested in how the trained model performs on new tasks, each of which is associated with a
>   dataset $\mathcal{D}$ specified for one task at a time. $\mathcal{D}$ thus has a support/context set and a query set. As also mentioned in the review, the applications of NPs to ML keep the task id inaccessible such that the model learns to infer it from the context. This is because what defines a new task in ML is just the context points.
>
> However, in CL with a replay memory, our training and testing set could both comprise samples from all the tasks. That is to say, the testing phase of CL does not have any new tasks, which is the fundamental difference from the ML problem setting. Therefore, we believe that it is valid that our training and test data use the same set of context points. Consequently, it is also valid for us to access the task IDs of the context set points to derive more informative task-specific priors.
>
> In practice, going beyond the ML setting, there have been several methods in the NP literature that use task id for *designating different parts of an NP model to different tasks* for multi-task learning [1,2] and *using the training samples as context at test time* for large-scale classification tasks [3,4].
>
> **Therefore, we believe that our use of NP for CL is valid and sincerely hope the reviewer reconsiders their rating.**
>
> - **Effect of context on the performance of NPCL**: To verify that the access to task ids make the task-specific context $\mathcal{C}^t$ more informative, we design a simple experiment. While training on an incremental task $t > 0$ onwards, after having derived the global distribution samples $z^G$,  we tinker with the flow of the task-specific context points $C^t$ to the different task-specific components of the NPCL. Namely, instead of directing $\mathcal{C}^t$ to its *correct* $t-th$ task encoder, we redirect it to the $j-th$ task encoder where, $j \neq t$ and $j$ are chosen at random from the pool of all seen task ids. Note that the presence of such randomly allocated context points during training implies that we now have a noisy task-specific prior, *i.e.,* $q_\phi(z^t$|$z^G, \mathcal{C}^j)$ to match in the second term of the right-hand side of our ELBO, *i.e.,* eq. 5 in the main paper.  We keep all our other training settings (including the loss coefficient values) unchanged.
>
>  Figure 4 in the global response PDF compares the performance of NPCL with the noisy task-specific priors on the two different memory sizes of S-CIFAR-10. While the performance gain of the NPCL over its noisy prior counterpart remains significant, we observe that in comparison with the ST-NPCL (which *lacks* hierarchy), the presence of noisy priors degrade the performance of NPCL (which *has* hierarchy) further as the replay memory size increases from 200 to 500. This is because, with a larger memory size, more context points from past tasks are diverted to the random task components during training. This leads to a noisier task-specific prior matching. Such noisy priors further lead to higher fluctuations in the accuracy, as marked by the larger standard deviations in their accuracy over ST-NPCL and NPCL. *This validates the fact that the performance of NPCL is connected to the conditioning on correct task-specific context and not merely on the higher capacity of the hierarchical latent variable model.*
>
> - **Line 110**:  In line 110, we mention optimizing parameters *on* $\mathcal{D}^t$ rather than *of* $\mathcal{D}^t$. Here, $\mathcal{D}^t$ refers to the training samples from task $t$ (building upon the notation from lines $93-94$ and $103$ in main paper). By joint optimization on $\mathcal{D}^t$ and the memory $\mathcal{M}$, we mean that if the entire parametric space of the model is allocated to learning $\mathcal{D}^t$ and $\mathcal{M}$, then interference from the future tasks is more likely.
>
>
> - **Superscripts in eq. 3**: As mentioned earlier, in a standard image classification setting like ours, the context points are not the only definition of a task given that the training and testing data in such settings can come from multiple tasks simultaneously. We thus use the superscripts $0:t$ to differentiate the task-specific context $C^{0:t}$ and target $X^{0:t}$. For the same reason, $z^t$ is conditioned on $C^t$.
>
> - **Line 151**: We are sorry for the confusion here. Conditioned on each $z \in $ {$z^G$}$_1^N$, we rely on sampling one set of {$z^t$}$_1^M$ from each task-specific NPCL distribution. Thus, for $M=1$, we have $N$ samples from each of the $t+1$ task-specific encoder, leading to a subtotal of $N \times (t + 1)$ latents.  Further, since $z^t$ for different tasks are sampled from distributions learned by different task encoders, this does not imply mapping every point to only one task. For a more clear explanation, we provide a rough sketch of our hierarchical sampling framework in Figure 3(b) of the attached PDF. Also, in Figure 3(a), we shed light on the effect of different $M$ and $N$ values on the accuracy of NPCL.
>
> **References**:
>
> [1] Kim, D., Cho, S., Lee, W., & Hong, S. (2022). Multi-Task Processes. *ICLR*.
>
> [2] Shen, J., Zhen, X., Worring, M., & Shao, L. (2021). Multi-Task Neural Processes. ArXiv, abs/2111.05820.
>
> [3] Wang, J., Lukasiewicz, T., Massiceti, D., Hu, X., Pavlovic, V., & Neophytou, A. (2022). NP-Match: When Neural Processes meet Semi-Supervised Learning. *ICML*.
>
> [4] Jung, M.C., Zhao, H., Dipnall, J.F., Gabbe, B.J., & Du, L. (2023). Multimodal Neural Processes for Uncertainty Estimation. ArXiv, abs/2304.01518.

---

> > ### Author Response · Authors · 2023-08-15
> >
> > Dear Reviewer v3Yx,
> >
> > We thank you again for taking your time reviewing this work. We put our best efforts to prepare the rebuttal to your questions. We would very much appreciate if you could engage with us with your feedback on our rebuttal. We would be glad to answer any further questions and clarify any concerns.
> >
> > Also, if you are satisfied with our answers, please consider revising your score.
> >
> > With best regards

---

> > > ### Comment · Reviewer_v3Yx · 2023-08-15
> > >
> > > Thank you for your answer. I'm still not sure I understand how the problem is modeled as a NP.
> > > In NPs the dataset is considered to be a dataset of different functions X->Y, where each sample is a specific function. NPs are trained as generative models of these functions in a way that enables flexible conditioning. So given a trained NP, we can generate a complete function given a context set of observed x,y points along the function, or even generate a new function without any observed (context) points.
> > > This formulation can be mapped to the few-shot learning setup (i.e. meta-learning), where each task is a function x->y, and again we have a dataset of those tasks (functions) and the few shot examples available for each task serve as the context of x,y pairs from the specific function.
> > > How is this formulation mapped to the continual learning setup? Does the equivalence between 'task' and 'function' hold here?
> > > From my understanding, in [1], the 'tasks' are different than the 'functions'. One way to think about it is that each function has a multi-dimension output y which corresponds to the different tasks.  Does this formulation apply to your model as well?

---

> > > > ### Author Response · Authors · 2023-08-16
> > > > **Further clarification on the problem setting**
> > > >
> > > > Dear Reviewer v3Yx,
> > > >
> > > > Thank you for the comment. We agree with you on NPs for the meta-learning (ML) settings including few shot problems. To clarify the problem setting and your further questions, we have prepared the following three points whilst being more specific in terms of notations.
> > > >
> > > > **A. ML vs CL (ours) setting:** In both ML and continual learning (CL) settings, we have multiple tasks and we would like to learn an NP to generate task-specific functions, which aligns with “NPs are trained as generative models of these functions in a way that enables flexible conditioning”. Specifically, for each task t, we have $Y = f_{Z_t}(X)$ and $Z_t$ is the latent variable conditioned on the task-specific contexts. As $Z_t$ is task-specific, this makes $f_{Z_t}$ also task-specific.
> > > >
> > > > There are two major differences between standard NPs for ML and our proposed NP for CL:
> > > >
> > > > 1. In standard NPs, $Z_t$ is conditioned on $(X^c_t, Y^c_t)$, which are context samples of task $t$. While in our NP, $Z_t$ is conditioned on $(X^c_t, Y^c_t, Z_G)$. $Z_G$ is from $X^c_G, Y^c_G$ which are the data samples from all the tasks. To accommodate the properties of CL better, we introduce a hierarchical model that introduces more sharing in $f_{Z_t}$ across the tasks via $Z_G$.
> > > > 2. The goals of ML and CL are quite different. ML aims to learn NPs that can generalize to new tasks. That is why NPs are tested in a new task $t*$ given its context samples $(X^c_t*, Y^c_t*)$. In CL, our goal is to have a good $f$ that can perform well on all the seen tasks. Therefore, there is no new task during inference. Similar to other works adapting NPs to learn such a good $f$ generalizing all the seen training tasks [1], our context samples during testing are a subset of the training data for the tasks seen so far.
> > > >
> > > > To summarize, CL requires to learn a general function $f$ that is task-specific but also performs well on all the seen tasks given a limited rehearsal memory. To achieve this, we leverage NP ability of generating task-specific functions with flexible conditioning. But instead of using the standard model of NPs, we propose a hierarchical model to introduce more sharing, tailored for the CL problem.
> > > >
> > > > **B. Task-to-function equivalence:** A CL task may or may not involve data points sampled from a single function $f_{Z_t}: x$->$y$. For a standard CL setup (like ours), the equivalence between a task and a function does hold since each data point $x$ in a task is mapped to a single label $y$. However, if we were to adapt NPs for a multi-label CL setting [3,4], then similar to [2], at each (incremental) training task, we would learn a function with a multi-dimensional output $y$.
> > > >
> > > > You are thus correct that the tasks and functions are not equivalent in [2]. But for our CL setup, where each $x$ has a single label $y$, a task–to–function equivalence holds. In fact, we note the difference in the nature of prediction variables $y$  between ours and [2] in the related work **lines 84-85 (see point (b))** in the main paper. Based on this discussion, we will rephrase the line to make it more explicit regarding the equivalence.
> > > >
> > > > **C. Limited context scenario:** Finally, just like the standard NPs, NPCL does not need a complete set of context points over all the learned functions to be able to infer these although a diverse set of context does certainly help. This is evident from our ablation on the varying context sizes during inference (lines 253-256 in main paper). Here, a context set size of as little as 5 still helps us maintain a competitive accuracy on S-CIFAR-10.
> > > >
> > > > We hope the above points help clarify your doubts regarding the usage of NPs in our problem setting. Please let us know if you have any further questions or concerns.
> > > >
> > > > We look forward to your feedback.
> > > >
> > > > *References*:
> > > >
> > > > [1] Wang, J., Lukasiewicz, T., Massiceti, D., Hu, X., Pavlovic, V., & Neophytou, A. (2022). NP-Match: When Neural Processes meet Semi-Supervised Learning. ICML.
> > > >
> > > > [2] Kim, D., Cho, S., Lee, W., & Hong, S. (2022). Multi-Task Processes. ICLR.
> > > >
> > > > [3] Liang, Y., & Li, W. (2022). Optimizing Class Distribution in Memory for Multi-Label Online Continual Learning. ArXiv, abs/2209.11469.
> > > >
> > > > [4] Du, K., Lyu, F., Li, L., Hu, F., Feng, W., Xu, F., Xi, X., & Cheng, H. (2022). Multi-Label Continual Learning using Augmented Graph Convolutional Network. ArXiv, abs/2211.14763.

---

> > > > > ### Comment · Reviewer_v3Yx · 2023-08-16
> > > > >
> > > > > Thanks, this is a bit clearer now, but I'm still not sure I understand the roles of $Z_G$ and the different $Z_t$'s.
> > > > > Could you please describe the way the model is used at test time?
> > > > > More specifically, given a new target $x$ coming from an unknown task $t^*$
> > > > > 1. What context is used to infer $Z_G$?
> > > > > 2. What context is used to infer the different $Z_t$'s?
> > > > > 3. Are the different $Z_t$'s inferred with the same encoder network? (i.e. are the weights shared).
> > > > > 4. How is the single predicted y computed?

---

> > > > > > ### Author Response · Authors · 2023-08-16
> > > > > > **Inference using NPCL**
> > > > > >
> > > > > > Dear reviewer v3Yx,
> > > > > >
> > > > > > We are happy to have made our problem setting more clear to you. Here is a brief summary of how NPCL performs task-agnostic CL inference for a given test sample $x*$.
> > > > > >
> > > > > > At test time, we first make use of the training samples stored in the CL rehearsal memory $\mathcal{M}$ as our context samples. Note that in addition to labels $y$ for the data points $x$, our setting also stores the task ID $t$ for each $x$ in the memory  $\mathcal{M}$. The availability of the latter task IDs help us derive the task-specific latents $Z_t$’s at test time similar to how we derive them during training.
> > > > > >
> > > > > > In what follows,for brevity, we conceal the role of the feature extractor $f$ (e.g., a ResNet-18) used to derive the features of a data point $x$ before feeding these to the NPCL.
> > > > > >
> > > > > > Let us say we are performing the inference after having seen $(t+1)$ tasks, i.e., we have trained a total of $(t+1)$ task-specific encoder modules of NPCL so far. The NPCL inference then proceeds with the following steps:
> > > > > >
> > > > > > - **Shared global encoder:** The context samples from  $\mathcal{M}$ are used to derive the $N$ Monte Carlo (MC) samples of global { ${Z_G}_{i=1}^N$ using the global encoder module. **Note that the global encoder is shared by all tasks.**
> > > > > >
> > > > > > - **Individual task encoders and their respective context:** For deriving $Z_t$'s, we rely on the task ID $t$ for each context sample $x \in   \mathcal{M}$. As shown in Fig. 2 in the main paper, the input to each task-specific encoder are the intra-task attended features $s^t$ concatenated with the $N$ global MC samples  ${Z_G}$. Following this, we sample $M$ task-specific latents {$Z_t$}$_{i=1}^M$ from the $(t+1)$ task encoders. Note that we have the access to the task ID $t$ for the memory samples (which as stated above, are simply a subset of the training samples used in a typical experience replay CL setup). **Also, note that the weights for the $t-$th task-specific encoder is shared only among the context samples belonging to the $t-$th task.**  Therefore, these task IDs can be used to redirect the features of the context samples to their respective task encoders. At the end of this step, we have a total of $(t+1) \times N \times M$ task-specific latents that we can possibly employ for decoding.
> > > > > >
> > > > > > - **Shared decoder:** Now, for a given task sample $x*$, we are left with $(t+1) \times N \times M$ number of possible task-specific latent $Z_t$ to infer from. What we know is that out of these $(t+1) \times N \times M$ task-specific latent samples, one set of $ N \times M$ latent samples belongs to the correct task encoder for $x*$, i.e., conditioning the decoding of $x*$ with this set of $ N \times M$ latent samples should give us the most accurate prediction $y*$. To identify this correct set, we first decode predictions for $x*$ conditioned on the entire set of $(t+1) \times N \times M$ of $Z_t$'s. In particular, we feed the concatenated feature [$x*; z_t$] to the decoder.
> > > > > >
> > > > > > - **Uncertainty-based inference:** The decoder module outputs a total of $(t+1) \times N \times M$ logits corresponding to each $Z_t$. Next, for each set of $ N \times M$ logits (corresponding to the individual task encoders), we compute the Shannon entropy scores using eq. (9) in the paper. We then consider the set of logits with the lowest Shannon entropy as the final predictions.
> > > > > > - **Single y** * **prediction:** To get the single final prediction $y*$, we compute the average of the above set of $ N \times M$ logits.
> > > > > >
> > > > > > We hope the above steps make the roles of $Z_G$ and $Z_t$’s more clear to you. We would be happy to answer any further questions. We look forward to your response.
> > > > > >
> > > > > > With best regards

---

> > > > > > > ### Author Response · Authors · 2023-08-21
> > > > > > >
> > > > > > > Dear Reviewer v3Yx,
> > > > > > >
> > > > > > > As we approach the end of the discussion period, we are thankful to you for your precious time and engagement.
> > > > > > >
> > > > > > > In particular, we were pleased to hear that our response made the validity of NPs in our setting more clear to you. We would thusly be grateful if you could reconsider your recommendation based on our follow-up discussion.
> > > > > > >
> > > > > > > Thank you again for your consideration.
> > > > > > >
> > > > > > > With best regards

---

> > > > > > > > ### Comment · Reviewer_v3Yx · 2023-08-22
> > > > > > > >
> > > > > > > > After the above discussion I accept that the presented model is a valid formulation of NPs. Since this addresses the main concern I had, I am raising my score.

---

### Official Review · Reviewer_fvgr · 2023-07-07

**Soundness:** 3 good
**Presentation:** 4 excellent
**Contribution:** 3 good
**Rating:** 7
**Confidence:** 4

**Summary:**

This paper presents an uncertainty-aware continual learning framework that utilizes Neural Processes (NPs). The NP model employs a hierarchical latent variable model in conjunction with an experience replay buffer, where a global latent variable captures inter-task correlations and task-specific latents encode more detailed knowledge. To prevent catastrophic forgetting, the method regularizes a Jenson-Shannon divergence between current and past distributions of latent variables. The paper employs entropy for uncertainty quantification.

**Strengths:**

- The concept of regularizing latent distributions between preceding and current tasks is both simple and intuitive.
- Experimental results demonstrate that the proposed approach is competitive with state-of-the-art continual learning methods, while also being capable of quantifying model uncertainty.
- The proposed method only stores two vectors for each task, showcasing memory efficiency in terms of experience replay.
- Generally, the paper is well-written and easy to understand.

**Weaknesses:**

- Given that the encoder includes self-attention and cross-attention layers, the proposed method exhibits quadratic complexity relative to the training data. This complexity could prove problematic when the training dataset is substantial (e.g., 10000).

**Questions:**

- Why was Jensen-Shannon Divergence (JSD) chosen as a regularization metric? Have other distribution divergence metrics, such as Kullback-Leibler Divergence (KLD), been explored?
- A key strength of the NP family is its few-shot learning capability. Given that the size of the replay buffer can be a limiting factor in some applications, would the proposed approach still be effective with a much smaller buffer size (e.g., 10)?
- How do accuracy and uncertainty depend on the size of the Monte Carlo sampling (N, M)?

**Limitations:**

The authors have addressed the limitations.

---

> ### Author Rebuttal · Authors · 2023-08-09
>
> We thank the reviewer for their careful reading and remarks on our work. In regard to the questions raised, we provide our explanations below:
>
> - **Choice of regularization metrics**: Thank you for your suggestion. We indeed consider KL-divergence as a possible alternative for our regularization metric. The table below reports the average accuracy of NPCL on S-CIFAR-10 (memory size = 200) and S-CIFAR-100 (memory size = 500) using JS and KL divergences as the regularization metrics. Overall, JS-divergence offers consistent gains over its asymmetric KL counterpart in both settings. This could be because KL-divergence induces harder constraints on the distributions of the global and task-specific posteriors in matching their respective past task priors. However, in CL, the evolving data distribution means that our global and task-specific posteriors keep changing as the model’s parameters adapt to fit the newly arrived data. Therefore, the symmetric JS-divergence, which has a relatively relaxed mid-point anchor between the current posterior and its past task prior to match, offers a better trade-off between the plasticity and stability of the model. Our observation is in line with previous work on NPs that use JS divergence as a superior choice for the prior matching loss term [1].
>
> |     NPCL          | S-CIFAR-10 | S-CIFAR-100 |
> |---------------|---------------------|----------------------|
> | JS-divergence | 63.78 +/- 1.7       | 37.43                |
> | KL-divergence | 62.55 +/- 1.58      | 35.82                |
>
> - **Few-shot replay setting**: Thanks for your suggestion. To evaluate the performance of NPCL in a few-shot replay setting, we report the accuracy of NPCL (ours), ER [2] and DER [3] on S-CIFAR-10 and S-CIFAR-100 with memory sizes of 5 and 10, respectively (see Fig. 2 in the global response PDF). Further, as pointed out by reviewer **qdzJ**, we also compare the expected calibration errors (ECE) of these models to study how well their predicted probabilities reflect the true likelihood of the labels.
>
> As shown in Figure 2 of the PDF in the global response, NPCL consistently outperforms ER and DER in both memory size settings in terms of accuracy (higher the better) and ECE (lower the better). Interestingly, on the memory size of 5 on S-CIFAR-100, we observe that while the ER outperforms the DER in terms of accuracy, the latter still offers more confident predictions characterized by a lower ECE than that of the ER. NPCL, on the other hand, still exhibits the lowest ECE in this setting. This observation depicts that on very small CL memory settings, while regularizing the predicted logits towards their old forms – as done by the DER – helps maintain better predictive confidence than the simple ER, regularizing the task distributions towards their old forms – as done by the NPCL – remains the superior way to enhance the model’s predictive confidence.
>
> - **Effect of Monte Carlo samples M and N**: Please refer to our global response alongside Figure 3(a) of the attached PDF.
>
> - **Quadratic complexity of NPCL relative to the training data**: This is correct. One possible solution for improving the quadratic complexity of the current implementation could be to replace the self-attention and cross-attention blocks as well as the global average pooling operations entirely with Multi-Head Attention Pooling (MAP) [4]. In particular, the inducing points variant of the MAP could be used to summarize the set of $n$ input points with $m$ inducing points where $m << n$. This way, the time complexity could be brought down from $\mathcal{O}(n^2)$ to $\mathcal{O}(nm)$ where $n$ is the size of the input context/target set and $m << n$ is the number of inducing points encoding the original $n$ points.
>
> We have currently performed a preliminary experiment with MAP where we replace only the global average pooling operations with MAP (please refer to our response to reviewer **1ufQ**). Per the advice from reviewer **1ufQ**,  the aforesaid experiment was done from the viewpoint of improving accuracy. We thus plan to explore MAPs for improving the computational complexity of NPs in CL as a potential direction for future work.
>
> **References**:
>
> [1] Wang, J., Lukasiewicz, T., Massiceti, D., Hu, X., Pavlovic, V., & Neophytou, A. (2022). NP-Match: When Neural Processes meet Semi-Supervised Learning. *ICML*.
>
> [2] Riemer, M., Cases, I., Ajemian, R., Liu, M., Rish, I., Tu, Y., & Tesauro, G. (2018). Learning to Learn without Forgetting By Maximizing Transfer and Minimizing Interference. *ICLR*.
>
> [3] Buzzega, P., Boschini, M., Porrello, A., Abati, D., & Calderara, S. (2020). Dark Experience for General Continual Learning: a Strong, Simple Baseline. *NeurIPS*.
>
> [4] Lee, J., Lee, Y., Kim, J., Kosiorek, A., Choi, S., & Teh, Y. W. (2019, May). Set transformer: A framework for attention-based permutation-invariant neural networks. In International conference on machine learning (pp. 3744-3753). *PMLR*.

---

> > ### Author Response · Authors · 2023-08-15
> >
> > Dear Reviewer fvgr,
> >
> > We thank you again for taking your time reviewing this work. We put our best efforts to prepare the rebuttal to your questions. We would very much appreciate if you could engage with us with your feedback on our rebuttal. We would be glad to answer any further questions and clarify any concerns.
> >
> > With best regards

---

### Official Review · Reviewer_qdzJ · 2023-07-07

**Soundness:** 3 good
**Presentation:** 3 good
**Contribution:** 2 fair
**Rating:** 6
**Confidence:** 3

**Summary:**

This paper introduces an uncertainty-aware continual learning framework based on neural processes. The proposed method casts CL into a hierarchical latent model from the global variables to the task-specific variables, and the corresponding regularization terms for each to ensure minimum forgetting along the course of new task learning. The naturally reliable uncertain estimation of NP facilitates class incremental setting by allowing accurate task-specific latent selections.

**Strengths:**

- The presentation of this paper is mostly clear, with the equations properly delivering the ideas.

- The idea of applying NP in CL, explicitly modeling task-specific latent variables, and uncertainty quantification for class-incremental CL are very intriguing.



**Weaknesses:**

**Writing and clarity**
The writing can be further improved.

For example, in my understanding, following Line 163, the two regularization terms are introduced to counter the forgetting instead of being 'two key aspects of forgetting in NPCL.'

In line 114, the authors mentioned 'uses generative factors', while the term 'generative factors' is not appropriately explained here or in the following sections.


**Quantitative comparisons**

While clear improvements over some baseline methods and other NP-based methods are reported, I believe the compared CL methods are all relatively classic but old. More up-to-date might be helpful to better position the proposed method.


Other minor points:

- Figure 2 is too small

- The authors might consider refining the formats of the reference list, as currently the formats are very inconsistent.



**Questions:**

Please see the Weakness section.

How is the uncertainty estimation of the proposed method quantitatively? Some further evaluation using metrics like ECE can be more insightful.

**Limitations:**

The authors discussed the limitations of the proposed method very comprehensively in Section 6.

---

> ### Author Rebuttal · Authors · 2023-08-08
>
> We thank the reviewer for their careful reading and remarks on our work. In regard to the questions raised, we provide our explanations below:
>
> - **Writing, figures and referencing formats**: Thank you for pointing these out. We will be incorporating these into our paper.
> - **CL Baselines**: Thank you for your suggestion. We agree that CL is a fast-moving field and many new works are emerging. The CL baselines opted by us have been frequently used in a range of other recent CL papers [1-4]. We thus use these clean and powerful baseline models to clearly validate the effectiveness of the proposed NP-based model in CL. We will expand the group of methods used in our comparison.
> - **Quantitative results for uncertainty**: Thank you for your suggestion. We agree with the idea of quantitative uncertainty comparisons, and following your suggestion, we report the calibration error of NPCL with other methods to compare how well their predicted probabilities reflect the true probabilities. In particular, we report the Expected Calibration Error (ECE) [5] and the more robust Adaptive Calibration Error (ACE) [6] of ER, DER, a baseline Attentive NP (ANP) [7], and NPCL (ours) on the S-CIFAR-10 and the S-CIFAR-100 settings with memory sizes 200 and 500, respectively (see *Figure 1* in the pdf attached with the *global* response).  ECE and ACE are widely-used metrics for uncertainty estimation. As shown in Figure 1 in the attached pdf, NPCL consistently produces the most well-calibrated output probabilities. Moreover, even the baseline ANP, whose accuracy scores are comparable with that of the ER (see Table 1 in the main paper), produces predictions with lower calibration errors than the ER method.
>
>
> **References**:
>
> [1] Kim, S., Noci, L., Orvieto, A., & Hofmann, T. (2023). Achieving a Better Stability-Plasticity Trade-off via Auxiliary Networks in Continual Learning. *CVPR*.
>
> [2] Sun, Z., Mu, Y., & Hua, G. (2023). Regularizing Second-Order Influences for Continual Learning. *CVPR*.
>
> [3] Boschini, M., Bonicelli, L., Buzzega, P., Porrello, A., & Calderara, S. (2022). Class-Incremental Continual Learning Into the eXtended DER-Verse. *IEEE TPAMI*, 45, 5497-5512.
>
> [4] Gong, D., Yan, Q., Liu, Y., Hengel, A.V., & Shi, J. (2022). Learning Bayesian Sparse Networks with Full Experience Replay for Continual Learning. *CVPR*, 109-118.
>
> [5] Guo, C., Pleiss, G., Sun, Y., & Weinberger, K.Q. (2017). On Calibration of Modern Neural Networks. *ICML*.
>
> [6] Nixon, J., Dusenberry, M.W., Zhang, L., Jerfel, G., & Tran, D. (2019). Measuring Calibration in Deep Learning. *CVPR Workshops*.
>
> [7] Kim, H., Mnih, A., Schwarz, J., Garnelo, M., Eslami, S.M., Rosenbaum, D., Vinyals, O., & Teh, Y.W. (2019). Attentive Neural Processes. *ICLR*.

---

> > ### Author Response · Authors · 2023-08-15
> >
> > Dear Reviewer qdzJ,
> >
> > We thank you again for taking your time reviewing this work. We put our best efforts to prepare the rebuttal to your questions. We would very much appreciate if you could engage with us with your feedback on our rebuttal. We would be glad to answer any further questions and clarify any concerns.
> >
> > Also, if you are satisfied with our answers, please consider revising your score.
> >
> > With best regards

---

### Author Rebuttal · Authors · 2023-08-08

We thank the reviewers for their careful comments and suggestions. Here we provide a single-page PDF that contains the figures and tables that we want the reviewers to see while considering our responses. We look forward to a helpful discussion period.

- **Effect of Monte Carlo samples on performance**: As asked by reviewers **fvgr** and **1ufQ**, we discuss the effect of the number of Monte Carlo (MC) samples for global *N* and task-specific latent variables *M* on accuracy during inference (please refer to the heatmap in Figure 3(a) of the attached PDF). In particular, we observe two favorable spots in terms of accuracy, one centered around $(M=1, N=50)$ and the other around $(M=10, N=20)$. It is worth noting that the total number of inference time MC samples grow quadratically with the number of tasks ‘t’, i.e., $\mathcal{O}$($N \times M \times t$), and that a higher number of samples leads to a larger computational overhead. For instance, based on eq. (9) in the main paper, the inference on the 10th task of S-CIFAR-100, i.e., $t = 10$, given the two favorable $M$ and $N$ values amounts to selecting the set of task-specific module predictions with the least uncertainty from a total of (a) $1 \times 50 \times 10 = 500$ predictions using $(M=1, N=50)$, and (b) $10 \times 20 \times 10 = 2000$ predictions using $(M=10, N=20)$. We, therefore, opt for the more efficient setting of $(M=1, N=50)$ throughout our experiments for all settings.

---

### Decision · Program_Chairs · 2023-09-21

**Decision:**

Accept (poster)

**Comment:**

The paper received mixed views. After considerable discussion between the authors and reviewers, most of the concerns of the reviewers have been addressed. The reviewer with the lowest score also mentioned that their concerns have been addressed by the rebuttal. In view of the consensus by the reviewers, it is recommended to accept the paper.

However, it is recommended that the authors update the paper to include the clarification of the settings as this was the main concern. While this has been addressed in the rebuttal, it would be required for a reader to understand the use of NP in the continual learning setting clearly and the task and function equivalence in this setting.